# Attack as Defense: Run-time Backdoor Implantation for Image Content Protection

## Abstract

As generative models achieve great success, tampering and modifying the sensitive image contents (i.e., human faces, artist signatures, commercial logos, etc.) have induced a significant threat with social impact. The backdoor attack is a method that implants vulnerabilities in a target model, which can be activated through a trigger. In this work, we innovatively prevent the abuse of image content modification by implanting the backdoor into image-editing models. Once the protected sensitive content on an image is modified by an editing model, the backdoor will be triggered, making the editing fail. Unlike traditional backdoor attacks that use data poisoning, to enable protection on individual images and eliminate the need for model training, we developed the first framework for run-time backdoor implantation, which is both time- and resource- efficient. We generate imperceptible perturbations on the images to inject the backdoor and define the protected area as the only backdoor trigger. Editing other unprotected insensitive areas will not trigger the backdoor, which minimizes the negative impact on legal image modifications. Evaluations with state-of-the-art image editing models show that our protective method can increase the CLIP-FID of generated images from 12.72 to 39.91, or reduce the SSIM from 0.503 to 0.167 when subjected to malicious editing. At the same time, our method exhibits minimal impact on benign editing, which demonstrates the efficacy of our proposed framework. The proposed run-time backdoor can also achieve effective protection on the latest diffusion models.

## 1 Introdcution

In recent years, advances in generative models have been remarkable, demonstrating exceptional performance in image-editing tasks. Numerous editing models and their variant applications, such as LaMa( Suvorov et al. (2022); Yu et al. (2023)) and Stable Diffusion( Rombach et al. (2022b); Lugmayr et al. (2022)), have been developed, expanding the capabilities of image modification. Image inpainting stands out as a typical task of image editing, which aims to repaint a specific area in the image. The image inpainting receives an image-mask pair and uses the mask as positional guidance to edit the image. Text-guided editing/inpainting receives an image-mask-text triple as input and repaints the mask area of image following the text instruction. As these image inpainting models are widely used, they can be maliciously employed for copyright logo erasuring, face replacement, background replacement, and other abusing and tampering of image content, which brings new challenges to the field of artificial intelligence security.

Backdoor attack( Liu et al. (2018)) is a well-known threat on deep neural networks, which has received more attention with the rise of generative models such as diffusion models and large language models Ramesh et al. (2021); Achiam et al. (2023). Attackers can implant a backdoor into deep learning models by poisoning training data or manipulating model weights Chen et al. (2023); Chou et al. (2023; 2024); Zhai et al. (2023) in the training stage, then activate the backdoor in the inference stage through a predefined trigger in the input. In this work, we innovatively leverage the backdoor of image-editing models as a defense to protect sensitive image content from being tampered. In other words, when an editing model is utilized to modify a protected region on the image, the model backdoor will be activated and the modification could fail.

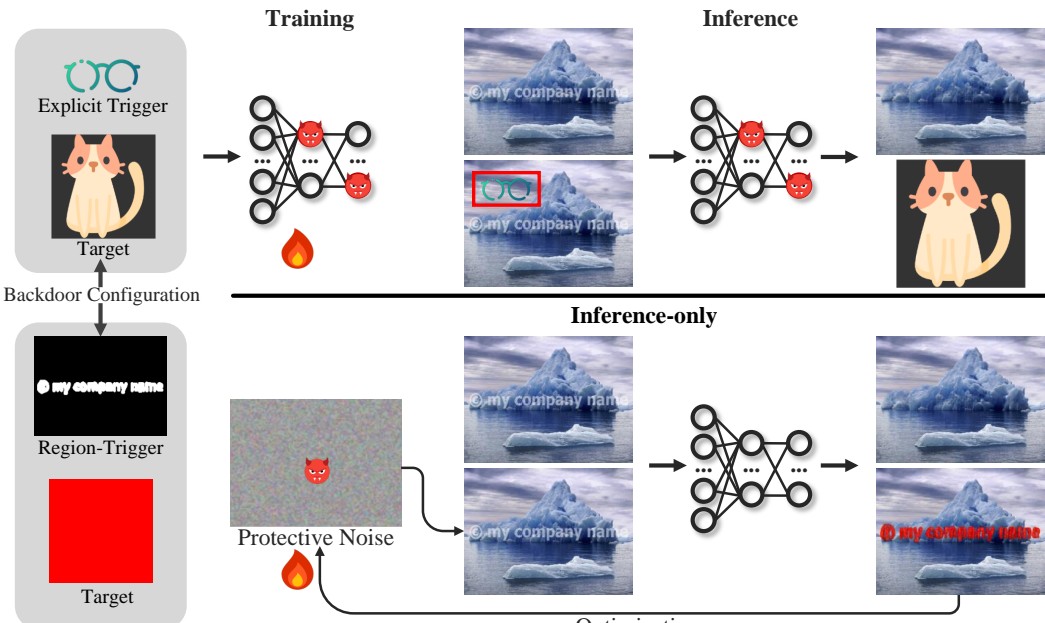

Figure 1: Paradigm comparison of traditional backdoor framework (top row) and the proposed Runtime implant framework (bottom row). The traditional approach requires obtaining a compromised model via Trojan training prior to deployment, with the backdoor being activated during the inference stage. In contrast, our runtime implant framework bypasses the need for prior poisoning, enabling the backdoor to be activated solely during inference. Conventional backdoor typically relies on explicit trigger, and our method leverages region-aware trigger, which is imperceptible and can be activated with editing location.

However, the unique characteristics of backdoor implantation present key challenges for this attack-as-defense process: as shown in the top row of Figure 1, ❶ it is difficult for photographers or creators to finetune or manipulate an image editing model, as attackers did before, to implant the backdoor. Even if it is possible, it is still hard to force others to use the released backdoor-ed model for image editing. ❷ The backdoor trigger on the image should be invisible, which can avoid disturbing the original image content created by the artists. ❸ Since the image editing operation could be diverse (e.g., various mask shapes and locations), backdoor activation must remain robust when the protected region is under modification. Additionally, editing other unprotected insensitive regions should be allowed to minimize the negative impact on legal modifications. To address these challenges, we propose our novel run-time backdoor implantation framework capable of implanting backdoors into image editing models without requiring model training. Figure 1 illustrates the overview of the attack framework, where the top row indicates the traditional backdoor implantation and the triggering process while the bottom row demonstrates ours. Specifically, for ❶, instead of fine-tuning the subject model, we generate protective noises that can induce inherent backdoors within the editing model when applied to the input images. For ❷, we establish the protected area on the image as the trigger for the backdoor, thus avoiding the risk of introducing visually perceptible disturbances, caused by explicit triggers, in the image content. To address ❸, we propose three collaborative optimization objectives to achieve region-aware backdoor activation. The implant loss is designed to activate the backdoor target when the edited area aligns with the protected region. The incomplete-trigger loss aims to activate the backdoor robustly when the edit area partially overlaps with the protected region. Additionally, the hide loss seeks to mitigate the impact of editing operations on other unprotected regions.

In summary, our contributions are as follows:

- We are the first to propose a run-time backdoor implantation framework for image models, enabling the injection of backdoors through protective noise during the inference stage, without requiring model retraining.

- We introduce the use of a protected area as a trigger mechanism to ensure minimal perceptible interference in the image, making the backdoor activation less detectable.

- We design a region-aware backdoor activation mechanism with three collaborative optimization objectives that allow the protective noise to reliably activate the backdoor across various image editing operations and remain minimal impact on legal image modifications.

- We evaluate the proposed framework on LaMa using inpainting datasets across six distinct scenarios. In the task of comprehensive inpainting, our approach reduces structural similarity index(SSIM) from 0.503 to 0.167, or improves CLIP-based fréchet inception distance(CLIP-FID) from 12.72 to 39.91. Furthermore, extended experiments demonstrate that our method effectively implants backdoors in diffusion models.

## 2 RELATED WORKS

**Image editing** involves the modification or enhancement of images using various models collectively known as image editing models. These models encompass tasks such as image inpainting Yu et al. (2023); Lugmayr et al. (2022); Yang et al. (2023); Corneanu et al. (2024), style transfer Kwon & Ye (2022); Zhang et al. (2023); Deng et al. (2022), and text-guided editing Tao et al. (2023); Nichol et al. (2021); Wang et al. (2023). Image inpainting is a key subtask focused on filling missing or corrupted regions of an image, and can be classified into two types: mask-guided inpainting and text-guided inpainting. In mask-guided inpainting Suvorov et al. (2022); Rombach et al. (2022b), models use an image-mask pair as input, where the mask defines the region to be reconstructed. The inpainting process relies on the surrounding unmasked regions to fill the masked area in a contextually coherent manner. In contrast, text-guided inpainting Manukyan et al. (2023); Ni et al. (2023); Xie et al. (2023) uses an image-mask-text triplet, where the missing region is filled based on both the surrounding image context and the textual input. Given the widespread use of inpainting-based image editing models, these functions can be exploited for tasks such as removing copyright logos, replacing faces, or altering backgrounds—raising significant concerns about image manipulation and misuse, which pose new challenges to AI security. Zhang et al. (2024) and Hu et al. (2023) are designed to ensure the authenticity of digital images by embedding imperceptible protection signals that can detect tampering and unauthorized modifications, but we want to prevent the tampering process itself. Although Salman et al. (2023) has performed adversarial learning on images to make them resistant to manipulation by diffusion models, it is not designed in the backdoor setting, where the protection will only be activated on specific conditions (i.e., triggers). Hence, it is not able to differentiate between malicious and benign editing operations, and could overreact on legal modifications. In light of this, we aim to strengthen protection against the misuse of fine grained image content (e.g. watermark, human face) while allowing for benign editing in authorized changeable areas (e.g., removing garbage in the background).

**Backdoor attack**( Liu et al. (2018); Hayase & Oh (2022); Qi et al. (2023)) is a classic threat to neural networks. Attackers usually implant backdoor into the model by data poisoning( Chen et al. (2017); Liu et al. (2020); Zhai et al. (2023)) during the training or fine-tuning phase, and then activate the backdoor by inputting samples containing trigger in the inference phase. In contrast to adversarial attacks( Cheng et al. (2024a;b)), which primarily exploit model vulnerabilities during inference, backdoor attacks are characterized by a predefined target orientation and require model training. Previous works( Chou et al. (2023); Sun et al. (2024); Chou et al. (2024); Li et al. (2024)) proposed to backdoor diffusion models by adding a special pattern as a trigger and train the model to make incorrect outputs when this trigger is encountered. However, since this modification is explicit to the model parameters, it makes the backdoor easily detectable. Yet, some backdoor detection methods( An et al. (2024; 2023); Hao et al. (2024)) for diffusion models have been proposed to detect the data distribution of models, especially on models released by third parties. Therefore, we propose a run-time backdoor framework to implant backdoor without manipulating the model, making it much more stealthy. We regard the position of the editing mask as the trigger. When protected area of the image is attempted to be edited, the backdoor is activated, leading to failed and unrealistic outputs.

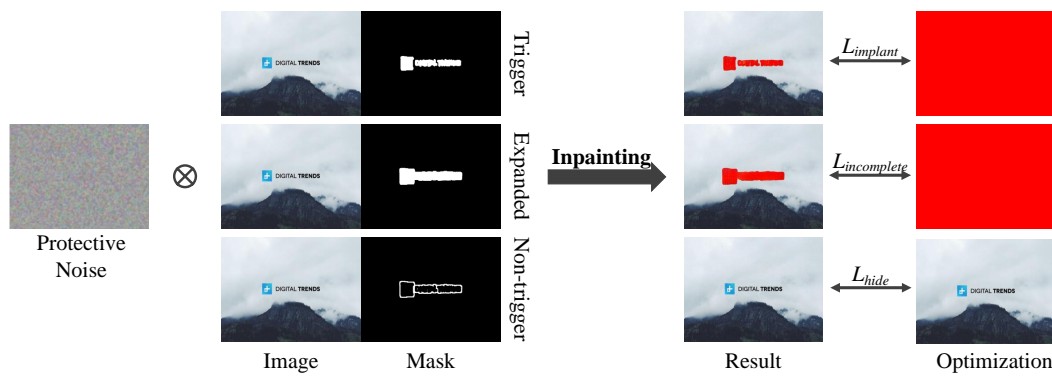

Figure 2: Optimization target of our run-time backdoor. We use three different edit regions as input to guide the optimization of protective noise. In the first row, the entire trigger region is employed to optimize the implant loss $\mathcal{L}_{implant}$. The second row utilizes an expanded trigger region to address incomplete activation loss $\mathcal{L}_{incomplete}$. The hide loss $\mathcal{L}_{hide}$ in third row applies editing to regions without trigger to minimize interference with benign modifications, thereby preserving the image's editability on non-trigger inputs.

## 3 METHOD

The overview of our proposed run-time backdoor framework is shown in Figure 1. In Section 3.1, we introduce the threat model and the scenario at first. Then, we describe the run-time backdoor implant framework in Section 3.2 and region-aware backdoor activation mechanism in Section 3.3.

### 3.1 THREAT MODEL

Image inpainting has gained widespread adoption as researchers increasingly release pre-trained image editing models to the public. In this work, we conceptualize three key parties involved in the process: The *developer* of inpainting models, responsible for training these models and making both the code and model weights available on platforms such as *hugging face*, allowing any user to download and utilize them. The *user*, who seeks to apply publicly available inpainting models and may download images shared on public platforms. The *defender*, representing image content protectors and copyright holders, such as artists and photographers, who aim to prevent unauthorized editing or misuse of their images.

In our proposed implant framework, the *developer* solely provides pre-trained inpainting models and does not engage in image editing or in the insertion of backdoors. The *user* may download images posted by content owners and employ the publicly available inpainting models for image manipulation. However, the user group may include malicious actors who exploit sensitive content for malicious purposes, such as illegal profiteering or infringement.

The *defender*, in turn, employs a run-time backdoor implantation mechanism in inpainting models, introducing protective noise perturbations into their images. A specific location within the image is designated as a trigger, producing a modified image that is visually indistinguishable from the original. The defender (the image owner) can then publicly post the protected image. If a malicious user attempts to edit the protected content using the inpainting model, the model will produce a distorted output, thus safeguarding the image from unauthorized manipulation.

### 3.2 RUN-TIME BACKDOOR IMPLANTATION

In this section, we formalize our *run-time backdoor* method. We describe the defense scenario using the image inpainting task, and the method can be easily transferred to applications of inpainting models such as Stable Diffuison( Rombach et al. (2022b)).

We first summarize the training paradigm of state-of-the-art image editing models, which can be described as follows: given an image editing model $\mathcal{IM}$, the inpainting function of $\mathcal{IM}$ is trained

with loss function:

$$\mathcal{L}_{\mathcal{IM}} = \alpha \cdot \mathcal{L}_{recon} + \beta \cdot \mathcal{L}_{adv} + \gamma \cdot \mathcal{L}_{perc}, \tag{1}$$

In this formulation, $\alpha$, $\beta$, and $\gamma$ serve as hyperparameters that balance the contributions of various components of the loss. Specifically, $\mathcal{L}_{adv}$ corresponds to the adversarial loss associated with the generative adversarial training process. The reconstruction loss, $\mathcal{L}_{recon}$, assesses the fidelity of the inpainted regions at the pixel level, while the perception loss, $\mathcal{L}_{perc}$, captures the differences in high-level features between the generated and target images, ensuring perceptual consistency. Specifically, the basic losses $\mathcal{L}_{recon}$ and $\mathcal{L}_{perc}$ for inpainting function can be jointly formalized as:

$$\mathcal{L}_{inpaint} = \mathbb{E}_{x,\mathcal{G}} \left[ \|x - \mathcal{G}(x)\|^2 \right], \tag{2}$$

where the $x$ and $\mathcal{G}(x)$ denote the input image and the generated result. Generally, traditional backdoor method requires to trojan the model $\mathcal{IM}$ by training the inpainting loss $\mathcal{L}_{inpaint}$ with an extra objective:

$$\mathcal{L}_{trojan} = \begin{cases} \mathbb{E}_{x \sim \mathcal{S},\mathcal{G}} \left[ \|x - \mathcal{G}(x)\|^2 \right], & \text{if } S = Clean \\ \mathbb{E}_{x \sim \mathcal{S},y,\mathcal{G}} \left[ \|y - \mathcal{G}(\mathcal{T}(x))\|^2 \right]. & \text{if } S = Poisonous \end{cases} \tag{3}$$

Here, $y$ represents the output target associated with the backdoor mechanism. The function $\mathcal{T}(x)$ denotes the image input containing the trigger, which is designed to activate the backdoor target. The training set distribution, denoted by $S$, consists of both clean data and poisoned data, reflecting a mixture of benign and manipulated samples used during model training.

However, the above paradigm of traditional backdoor is not suitable for our attack-as-defense scenario. Training a trojan model is both time- and resource-expensive for defenders. Our run-time backdoor is completely based on released models. Given an inpainting model $\mathcal{IM}$, it receives an image $x$ and a mask $m$ as input, and returns an image $r$ as the repainted result. This step can be written as $r = \mathcal{IM}(x, m)$. Our objective is to train a set of protective perturbations, denoted as $\mathcal{P}$. These perturbations are applied to the original image such that $\mathcal{P}(x) = x \otimes \mathcal{P}$. We use perturbations of the same shape as the image to achieve protection. The resulting perturbed image, $\mathcal{P}(x)$, is designed to resist malicious modification attempts while remaining visually indistinguishable from the original image to the human observer.

We achieve this by optimizing the two fundamental objectives $\mathcal{L}_{implant}$ and $\mathcal{L}_{hide}$:

$$\mathcal{L}_{implant} = \mathbb{E}_{\phi,\mathcal{P}(x),m} \left[ \|\phi_x - \mathcal{IM}(\mathcal{P}(x), \mathcal{T}(m))\|^2 \right], \tag{4}$$

$$\mathcal{L}_{hide} = \mathbb{E}_{\mathcal{P}(x),m} \left[ \|\mathcal{IM}(x, m_0) - \mathcal{IM}(\mathcal{P}(x), m_0)\|^2 \right], \tag{5}$$

We designate the region of the protected content within the image as the trigger. The backdoor target $\phi_x$ is generated specifically for each input sample. Detailed implementation of backdoor target can be found in Appendix B. The mask $\mathcal{T}(m)$ here is the entire protected region and we force the model to returns a distorted image similar to target $\phi_x$. The implant loss, denoted as $\mathcal{L}_{implant}$, is designed to optimize the perturbation such that $\mathcal{P}(x)$ effectively activates the backdoor in the presence of a trigger $m$. The hide loss, $\mathcal{L}_{hide}$, ensures operations outside the trigger region produce results similar to those from the original image $x$. The mask $m_0$ used in $\mathcal{L}_{hide}$ is specifically defined to exclude any overlap with the protected region, thereby ensuring benign editing behavior in those regions.

## 3.3 REGION-AWARE BACKDOOR ACTIVATION MECHANISM

The loss optimization procedure is illustrated in Figure 2. Through the optimization of the implant loss $\mathcal{L}_{implant}$, the core backdoor mechanism is implanted at run-time via the introduction of protective noise. However, in certain cases, malicious users do not modify the entire sensitive area, and malicious manipulation may cover only a portion of the trigger region. In these instances, the input trigger is incomplete, we still anticipate the successful activation of the backdoor mechanism. Based on this, we expand the mask region used in training perturbations as a potential protection region. We design a mask expansion strategy $\mathcal{E}(\cdot)$, which generates a new mask $\mathcal{E}(m)$ to expand the masked region in the image. We conduct our incomplete activation loss $\mathcal{L}_{incomplete}$ as:

$$\mathcal{L}_{incomplete} = \mathbb{E}_{\phi,\mathcal{P}(x),m} \left[ \|\phi_x - \mathcal{IM}(\mathcal{P}(x), \mathcal{E}(m))\|^2 \right], \tag{6}$$

$$\mathcal{E}(m) = conv2d(m, size_{kernel}, size_{padding}), \tag{7}$$

given that the mask image is binary, the value in the masked (white) region is 1, while the rest is 0. Thus, the operation $\mathcal{E}(\cdot)$ expands the white region through convolution. By optimizing Equation (6), the backdoor can be activated even when trigger is incomplete. However, a corresponding challenge is emerged: the backdoor may also be activated when editing regions outside the trigger region (e.g., performing benign erasure in the background), as shown in the yellow box in Figure 4. To reduce the impact of editing on non-trigger regions of the image, we propose to jointly optimize $\mathcal{L}_{hide}$:

$$\mathcal{L}_{hide} = \mathbb{E}_{\mathcal{P}(x),m} \left[ \| \mathcal{IM}(x, \mathcal{E}'(m)) - \mathcal{IM}(\mathcal{P}(x), \mathcal{E}'(m)) \|^2 \right], \tag{8}$$

$$\mathcal{E}'(m) = max(0, min(1, \mathcal{E}(m) - \mathcal{T}(m))), \tag{9}$$

By optimizing Equation (8), the trigger region in image can be effectively distinguished by model, preventing the backdoor from being mis-activated when editing does not involve the protected area. We optimizing above objectives in parallel. The total loss function is:

$$\mathcal{L}_{total} = \mathcal{L}_{implant} + \mathcal{L}_{incomplete} + \mathcal{L}_{hide}. \tag{10}$$

## 4 EVALUATION

In this section, we evaluate our Run-time Backdoor on image inpainting task at first. Then, we present ablation studies of loss functions and perturbation bounds. We discuss transferability of our framework in the end.

### 4.1 EXPERIMENT SETUP

**Model & Scenario Selection**. In our evaluation, we utilized two state-of-the-art image inpainting models: LaMa( Suvorov et al. (2022)) and MAT( Li et al. (2022)). These models represent convolutional neural network architecture and transformer architecture, respectively. Additionally, we included Latent Diffusion( Rombach et al. (2022b)) as a comparison for diffusion-based inpainting paradigm. To assess the performance of resistance to editing, we employed subsets from four datasets with different scenarios: Places2( Zhou et al. (2017)), CelebA-HQ( Karras (2017)), Watermark, and Car Brands Images. Each subset consisting of 50 randomly selected images. CelebA-HQ is an enhanced version of the CelebA dataset, consisting of high-resolution images of celebrities, we apply the $256 \times 256$ image sizes. Places2 is a comprehensive dataset comprising over 400 scene categories, we apply the $512 \times 512$ image sizes. Watermark dataset including 30 image-mask pairs. We collected released watermark-mask pairs from Kaggle and supplemented it with another watermark dataset with manually annotated masks. The image sizes include $768 \times 1024$ and $768 \times 768$. Car Brands Images consists of a collection of car images with logos.

**Implant Settings**. We define the region in the image corresponding to the mask to serve as trigger. For the trigger region of CelebA-HQ( Karras (2017)), and Places2( Zhou et al. (2017)) datasets, we apply standard $64 \times 64$ centering mask. We apply manually annotated mask on Watermark dataset and Car Brands Images, the masks locate at the regions of watermark and car logo respectively. Subsequently, we randomly retain half of the expanded mask region generated by Equation (7) to act as the incomplete trigger. We use the incomplete trigger to subtract the part that intersects with the trigger mask to get a mask without trigger. We empirically set the weight of the $\mathcal{L}_{hide}$ terms to 2, and the number of training iterations for the protective noise to 20. We adopt pure color image or inverted color image as the target, the detail will be provided in detail in the Appendix B.

**Metric Settings**. Given the lack of prior research on the run-time backdoor paradigm in image inpainting tasks, there is no established baseline for direct comparison. To address this, we employ several metrics to assess the resistance of implanted image to malicious editing. Specifically, we use Contrastive Language-Image Pretraining FID (CLIP-FID)( Radford et al. (2021)) and Learned Perceptual Image Patch Similarity (LPIPS)( Zhang et al. (2018)) to quantify the semantic plausibility of the edited image, where higher values indicate greater distortion. The Structural Similarity Index Measure (SSIM)( Wang et al. (2004)) is employed to evaluate the degradation in fidelity of local structure and texture, where lower values signify greater distortion. Additionally, we use Peak Signal-to-Noise Ratio (PSNR) to assess the loss of the refinement and restoration quality in the image, with lower scores indicating higher distortion. CLIP-FID is calculated on the entire image to

Figure 3: Examples illustrating the qualitative resistance of implanted imge to editing. The red-circled area in the figure highlights the inpainting result.

Table 1: Resistance performance to editing of the run-time backdoor framework

| Datasets | Editing Region | Input | LaMa | | | | MAT | | | |
|---|---|---|---|---|---|---|---|---|---|---|
| | | | CLIP-FID↑ | LPIPS↑ | SSIM↓ | PSNR↓ | CLIP-FID↑ | LPIPS↑ | SSIM↓ | PSNR↓ |
| CelebA-HQ | Trigger | Imp. | 39.91 | 0.083 | 0.167 | 8.98 | 23.92 | 0.083 | 0.294 | 9.56 |
| | | Ben. | 12.72 | 0.017 | 0.503 | 18.24 | 2.12 | 0.008 | 0.631 | 22.57 |
| | | Diff. | +27.18 | +0.066 | -0.336 | -9.26 | +21.80 | +0.075 | -0.337 | -13.01 |
| | Incmp. | Imp. | 23.39 | 0.047 | 0.277 | 13.31 | 9.46 | 0.039 | 0.398 | 15.05 |
| | | Ben. | 3.25 | 0.001 | 0.556 | 19.93 | 1.59 | 0.006 | 0.653 | 22.78 |
| | | Diff. | +20.14 | +0.046 | -0.279 | -6.62 | +7.86 | +0.033 | -0.255 | -7.74 |
| | W/o. | Imp. | 0.845 | 0.004 | 0.496 | 22.55 | 0.61 | 0.003 | 0.585 | 23.98 |
| | | Ben. | 0.610 | 0.003 | 0.556 | 24.26 | 0.47 | 0.002 | 0.653 | 25.89 |
| | | Diff. | +0.235 | +0.001 | -0.059 | -1.71 | +0.14 | +0.001 | -0.132 | -1.91 |
| Places2 | Trigger | Imp. | 11.40 | 0.075 | 0.099 | 8.46 | 29.37 | 0.073 | 0.219 | 10.04 |
| | | Ben. | 2.88 | 0.029 | 0.375 | 15.81 | 24.73 | 0.083 | 0.238 | 8.08 |
| | | Diff. | +8.52 | +0.046 | -0.276 | -7.35 | 4.63 | -0.010 | -0.019 | 1.95 |
| | Incmp. | Imp. | 5.06 | 0.039 | 0.202 | 12.52 | 2.50 | 0.029 | 0.342 | 15.82 |
| | | Ben. | 1.94 | 0.019 | 0.401 | 16.55 | 1.19 | 0.023 | 0.449 | 15.81 |
| | | Diff. | +3.12 | +0.02 | -0.199 | -4.03 | +1.31 | +0.007 | -0.107 | +0.01 |
| | W/o. | Imp. | 0.63 | 0.008 | 0.408 | 19.74 | 0.22 | 0.003 | 0.612 | 24.00 |
| | | Ben. | 0.49 | 0.005 | 0.401 | 20.98 | 0.11 | 0.002 | 0.449 | 25.51 |
| | | Diff. | +0.14 | +0.003 | +0.007 | -1.24 | +0.11 | +0.001 | -0.151 | -1.51 |
| Car Brands | Trigger | Imp. | 22.62 | 0.141 | 0.091 | 6.14 | 17.93 | 0.064 | 0.296 | 9.81 |
| | | Ben. | 8.65 | 0.047 | 0.362 | 13.26 | 14.16 | 0.063 | 0.312 | 8.51 |
| | | Diff. | +13.97 | +0.094 | -0.271 | -7.12 | +3.77 | +0.001 | -0.017 | +1.29 |
| | Incmp. | Imp. | 9.39 | 0.069 | 0.168 | 11.45 | 4.34 | 0.026 | 0.324 | 14.27 |
| | | Ben. | 3.34 | 0.024 | 0.421 | 15.58 | 3.25 | 0.022 | 0.434 | 14.56 |
| | | Diff. | +6.05 | +0.045 | -0.253 | -4.13 | +1.09 | +0.004 | -0.110 | -0.29 |
| | W/o. | Imp. | 0.594 | 0.008 | 0.340 | 18.83 | 0.11 | 0.003 | 0.599 | 23.38 |
| | | Ben. | 0.259 | 0.004 | 0.421 | 20.74 | 0.08 | 0.002 | 0.434 | 24.59 |
| | | Diff. | +0.335 | +0.004 | -0.081 | -1.91 | +0.03 | +0.001 | -0.165 | -1.21 |
| Watermark | Trigger | Imp. | 35.60 | 0.108 | 0.173 | 7.74 | 31.95 | 0.069 | 0.268 | 11.29 |
| | | Ben. | 26.44 | 0.057 | 0.459 | 15.41 | 27.14 | 0.061 | 0.421 | 16.07 |
| | | Diff. | +9.16 | +0.050 | -0.286 | -7.67 | +4.80 | +0.008 | -0.153 | -4.77 |
| | Incmp. | Imp. | 18.15 | 0.061 | 0.221 | 10.02 | 17.21 | 0.037 | 0.309 | 14.14 |
| | | Ben. | 12.44 | 0.031 | 0.523 | 16.21 | 13.31 | 0.033 | 0.476 | 16.64 |
| | | Diff. | +5.71 | +0.030 | -0.302 | -6.19 | +3.89 | +0.004 | -0.167 | -2.49 |
| | W/o. | Imp. | 0.11 | 0.004 | 0.454 | 26.62 | 0.16 | 0.004 | 0.522 | 27.45 |
| | | Ben. | 0.05 | 0.001 | 0.523 | 32.26 | 0.08 | 0.002 | 0.476 | 32.14 |
| | | Diff. | +0.06 | +0.003 | -0.069 | -5.64 | +0.08 | +0.002 | -0.046 | -4.69 |

measure the global distortion, and the remaining metrics are calculated on the edited region to reflect the local distortion.

## 4.2 RESISTANCE PERFORMANCE TO EDITING

We conducted a comparative analysis of the editing resistance of run-time backdoor implant method in the LaMa( Suvorov et al. (2022)) and MAT( Li et al. (2022)) models across four scenarios. We report the average distortion levels for each dataset, utilizing four inpainting iterations for each input sample to derive the metrics presented in Table 1. The metrics evaluate the discrepancies in editing results between benign images and those embedded with perturbations across three different modification regions. Table 1 outlines these findings, with the first column indicating the datasets.

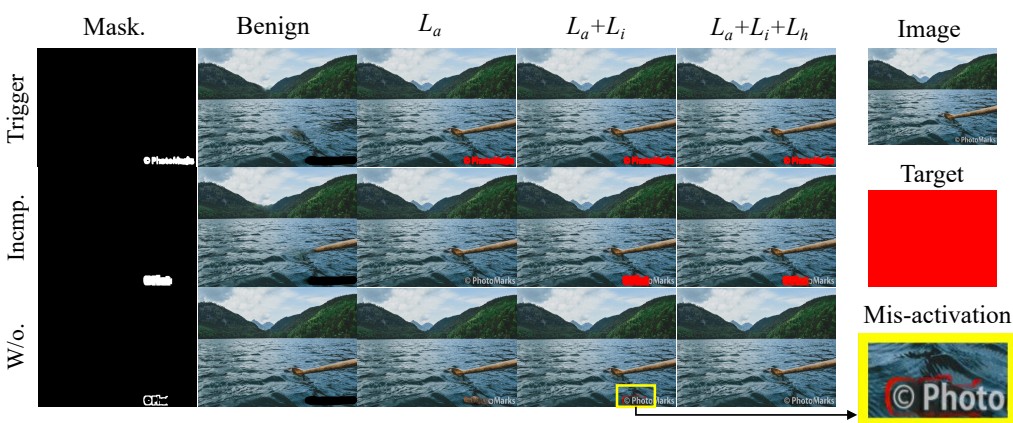

Figure 4: Example of qualitative ablation study on loss functions.

Table 2: Ablation study on loss functions.

| Dataset | Loss $\mathcal{L}_a$ | $\mathcal{L}_i$ | $\mathcal{L}_h$ | CLIP-FID↑ Trigger | Incmp. | W/o.↓ | LPIPS↑ Trigger | Incmp. | W/o.↓ | SSIM↓ Trigger | Incmp. | W/o.↑ | PSNR↓ Trigger | Incmp. | W/o.↑ |
|---|---|---|---|---|---|---|---|---|---|---|---|---|---|---|---|
| CelebA-HQ | Benign | | | 12.72 | 3.26 | 0.612 | 0.017 | 0.010 | 0.003 | 0.503 | 0.556 | 0.556 | 18.24 | 19.93 | 24.26 |
| | ✓ | | | 36.50 | 9.071 | 0.867 | 0.098 | 0.021 | 0.005 | 0.175 | 0.368 | 0.405 | 8.61 | 17.57 | 21.82 |
| | ✓ | ✓ | | 36.01 | 22.53 | 4.48 | 0.098 | 0.052 | 0.012 | 0.158 | 0.265 | 0.324 | 8.69 | 12.84 | 19.33 |
| | ✓ | ✓ | ✓ | 39.91 | 23.39 | 0.845 | 0.083 | 0.047 | 0.004 | 0.167 | 0.277 | 0.496 | 8.98 | 13.31 | 22.55 |
| Place2 | Benign | | | 2.87 | 1.94 | 0.043 | 0.029 | 0.019 | 0.005 | 0.375 | 0.401 | 0.401 | 15.81 | 16.55 | 20.98 |
| | ✓ | | | 10.28 | 2.84 | 0.592 | 0.083 | 0.028 | 0.007 | 0.072 | 0.271 | 0.364 | 8.09 | 15.22 | 19.22 |
| | ✓ | ✓ | | 10.84 | 5.54 | 1.59 | 0.084 | 0.048 | 0.013 | 0.074 | 0.175 | 0.273 | 8.28 | 11.27 | 16.63 |
| | ✓ | ✓ | ✓ | 11.40 | 5.06 | 0.625 | 0.075 | 0.039 | 0.007 | 0.099 | 0.202 | 0.408 | 8.46 | 12.52 | 19.74 |

The second column delineates the operational regions: "Trigger" refers to the editing of the trigger region, "Incmp." indicates the intersection of the editing area with the trigger, and "W/o." signifies the editing of non-trigger regions. The third column distinguishes between the input images, with "Ben." representing benign images and "Imp." indicating images post-perturbation. The subsequent columns present the resistance performance to editing of the run-time backdoor against various image editing models, evaluated when the perturbation bound is set at $6/255$.

As illustrated in Table 1, the run-time implantation demonstrates robust resistance performance to editing across various scenarios when applied to editing operations within protected regions. Run–time implantation causes significant damage to the structural similarity metrics and semantic consistency of image editing results. In comprehensive scenarios with Places2( Zhou et al. (2017)), our approach reduces the structural similarity metrics(SSIM) by 0.336. In the context of facial editing using the CelebA-HQ( Karras (2017)) dataset, global coherence (CLIP-FID) exhibited a minimum performance damage of 21.80, with the local coherence in the editing region (LPIPS) by 0.066. In cases where the editing on the region without trigger (W/o.), the differences between editing results of images with perturbations(Imp.) and those of original images(Ben.) are minimal. Moreover, the metrics in the rows of "W/o." indicating that the perturbation has a negligible compromise the visual quality or realism of the edited images.

As shown in Figure 3, we present some qualitative results on Watermark dataset to analyze the resistance performance of our run-time backdoor to malicious editing. The red-circled area in the figure highlights the inpainting result. The original image exhibits minimal resistance to watermark removal, whereas the implanted image demonstrates significantly enhanced protection for the watermark region. More qualitative results can be found in Appendix C. These results highlight the effectiveness of our run-time backdoor framework in maintaining resistance across different editing region, ensuring minimal compromise of the protected regions under subtle perturbation conditions.

## 4.3 ABLATION STUDIES

**Loss function ablation**. Table 2 presents the ablation study on different loss functions, where $\mathcal{L}_a$, $\mathcal{L}_i$ and $\mathcal{L}_h$ represent implant loss $\mathcal{L}_{implant}$, incomplete activation loss $\mathcal{L}_{incomplete}$ and hide loss $\mathcal{L}_{hide}$, respectively. As shown in the first column of Table 2, the three loss functions are progressively

Figure 5: Example of qualitative ablation study on perturbation bound.

Table 3: Ablation study on perturbation bound.

| Dataset | Perturbation Bound | CLIP-FID↑ Trigger | Incmp. | W/o.↓ | LPIPS↑ Trigger | Incmp. | W/o.↓ | SSIM↓ Trigger | Incmp. | W/o.↑ | PSNR↓ Trigger | Incmp. | W/o.↑ |
|---|---|---|---|---|---|---|---|---|---|---|---|---|---|
| CelebA-HQ | Benign | 12.72 | 3.26 | 0.612 | 0.017 | 0.010 | 0.003 | 0.503 | 0.556 | 0.556 | 18.24 | 19.93 | 24.26 |
| | $\ell_\infty = 2/255$ | 38.08 | 12.70 | 0.604 | 0.048 | 0.018 | 0.003 | 0.234 | 0.441 | 0.702 | 11.12 | 16.80 | 24.18 |
| | $\ell_\infty = 3/255$ | 38.08 | 18.80 | 0.642 | 0.064 | 0.031 | 0.003 | 0.223 | 0.377 | 0.660 | 9.97 | 14.81 | 23.70 |
| | $\ell_\infty = 6/255$ | 39.91 | 23.39 | 0.845 | 0.083 | 0.047 | 0.004 | 0.167 | 0.277 | 0.496 | 8.98 | 13.31 | 22.55 |
| | $\ell_\infty = 13/255$ | 38.81 | 22.10 | 0.996 | 0.098 | 0.054 | 0.006 | 0.107 | 0.194 | 0.346 | 9.15 | 12.86 | 20.94 |
| Place2 | Benign | 2.87 | 1.94 | 0.485 | 0.029 | 0.019 | 0.004 | 0.375 | 0.401 | 0.401 | 15.81 | 16.55 | 20.98 |
| | $\ell_\infty = 2/255$ | 7.26 | 2.75 | 0.594 | 0.050 | 0.025 | 0.004 | 0.215 | 0.329 | 0.547 | 10.51 | 14.80 | 20.69 |
| | $\ell_\infty = 3/255$ | 9.00 | 4.74 | 0.608 | 0.060 | 0.032 | 0.005 | 0.167 | 0.278 | 0.512 | 9.61 | 13.70 | 20.46 |
| | $\ell_\infty = 6/255$ | 11.40 | 5.06 | 0.625 | 0.075 | 0.039 | 0.007 | 0.099 | 0.202 | 0.408 | 8.46 | 12.52 | 19.74 |
| | $\ell_\infty = 13/255$ | 10.61 | 4.81 | 0.792 | 0.080 | 0.044 | 0.008 | 0.044 | 0.143 | 0.321 | 8.44 | 12.04 | 18.63 |

incorporated into the optimization process. Figure 4 presents the qualitative results of the ablation study on the three loss functions. The first column represents the input masks, with the first one corresponding precisely to the trigger region. The second column shows the inpainting result of the original image, while the subsequent three columns display the inpainting results of images with noise implanted using different loss functions.

The results in Table 2 and Figure 4 indicate that when an incomplete trigger (Incmp.) is introduced, the activation of the backdoor is significantly enhanced after optimizing $\mathcal{L}_i$. For instance, in the comprehensive scene Places2 Zhou et al. (2017), the semantic rationality distortion index CLIP-FID improves from 2.84 to 5.54 (compared to 1.94 for the original image) after optimizing $\mathcal{L}_i$, while LPIPS increases from 0.028 to 0.048 (compared to 0.019 for the original image). Additionally, without optimizing hide loss $\mathcal{L}_h$, the editing results (W/o.) in regions without trigger exhibit significant distortion. The yellow box in the Figure 4 illustrates that when optimizing only for implant loss and incomplete loss, editing to non-trigger regions may still mis-activate some backdoor targets. Consequently, it is essential to optimize the $\mathcal{L}_h$, to mitigate false activations of backdoors and minimize interference with benign edits. In terms of structural rationality, the SSIM index is highest when $\mathcal{L}_h$ is optimized in both scenarios. Remarkably, in the Places2 scenario of Table 1, the SSIM of the image's editing result after perturbation implantation even surpasses that of the original image (0.408 vs. 0.401).

**Perturbation bound ablation**. Table 3 provides a quantitative assessment of the impact of varying perturbation bounds of protective noise on resistance of editing performance. Specifically, perturbation bounds were set at $1/255$, $3/255$, and $6/255$, respectively. As the perturbation bound increases, the distortion in the resulting image edits becomes more pronounced, with more evident alterations in the visual appearance of the image. When the perturbation bound increases from $6/255$ to $13/255$, the distortion in the image editing results remains comparable is close, with the latter not consistently showing improvement. This suggests that increasing the perturbation bound does not necessarily lead to better performance and may even plateau or diminish in effectiveness. At a perturbation bound of $13/255$, noticeable pixel-level changes begin to emerge.

Moreover, the "W/o." column of Table 3 indicates reducing the perturbation bound can mitigate the influence of the editing operation in unprotected areas, though this reduction leads to weaker protection in more sensitive regions. For instance, under a perturbation bound of $1/255$, the fidelity of

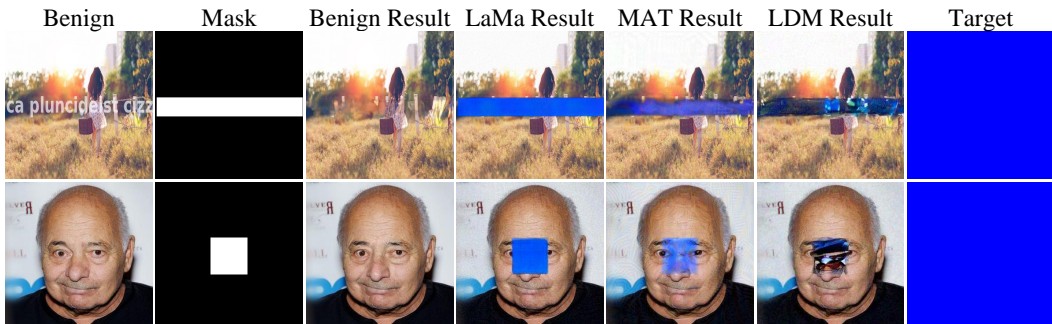

| Benign | Mask | Benign Result | LaMa Result | MAT Result | LDM Result | Target |

Figure 6: Qualitative example of the resistance performance of editing on different models.

the edited images in the "W/o." condition appears superior; however, qualitative analysis in Figure 5 indicates that this bound renders the backdoor mechanism largely ineffective. At a perturbation bound of $3/255$, edits involving regions containing the trigger can successfully activate the backdoor mechanism, although activation becomes unreliable when the trigger is incomplete.

Therefore, it is advisable to increase the noise intensity cautiously, ensuring that it does not introduce visible pixel changes. Empirical observations suggest that a perturbation bound of $3/255$ strikes an appropriate balance, offering sufficient protection without perceptible distortion. Detailed selection criteria can be found in Appendix E

### 4.4 TRANSFERABILITY

We compare the inpainting performance of different models in Figure 6, demonstrating the effectiveness of the run-time backdoor injection framework across various architectures. The latent diffusion model (LDM) is tested with input images of size $256 \times 256$ due to computational limitations. The analysis shows that LaMa( Suvorov et al. (2022)), a convolutional model, produces outputs closest to the target when using a solid color as the target, followed by MAT( Li et al. (2022)), a transformer-based model, which maintains better global structure due to its attention mechanism. LDM( Rombach et al. (2022b)), due to its denoising process, generates less accurate results, often producing unwanted blue pixels. perturbation bounds for LaMa and MAT were set at $6/255$, while LDM used a higher amplitude of $13/255$. Furthermore, our implantation framework can keep the backdoor effective even after data augmentation operations such as flipping, cropping, or rescaling. Further details on these parameters are discussed in the Appendix D.

## 5 CONCLUSION

We propose a resource-efficient run-time implant framework designed to expose pre-existing backdoors in models while minimizing both time and space consumption. This framework only requires learning protective noise for images during the inference phase and activates the backdoor through a trigger based on the editing region. It has been thoroughly evaluated across various image inpainting models, demonstrating its effectiveness in distorting editing operations when the protected region is involved. The results show that our implanted protective noise can significantly degrade restoration performance, increasing the CLIP-FID score from 12.72 to 39.90, or reducing the SSIM of the generated content from 0.503 to 0.167 on average.

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

# Appendix

# Attack as Defense: Run-time Backdoor Implantation for Image Content Protection

## A    Detailed Implementation of Image Inpainting

In image inpainting models, an image-mask pair $\langle x, m \rangle$ is typically provided as input, In the mask $m$, the masked area is represented as white, with a corresponding value of 1, while the non-masked area is represented as black, with a corresponding value of 0.

$$m(i,j) = \begin{cases} 1, & if \quad (i,j) \quad is \quad in \quad the \quad mask \quad region \quad (white) \\ 0, & if \quad (i,j) \quad is \quad outside \quad the \quad mask \quad region \quad (black) \end{cases} \tag{11}$$

This binary mask is used to delineate the regions for inpainting, where the white areas (value 1) guide the model to predict and fill the missing content, and the black areas (value 0) preserve the original image content. The term $r$ is used to denote the final outcome image, while $\mathcal{G}(x)$ refers to the content predicted by the inpainting model. The relationship between $r$ and $\mathcal{G}(x)$ can be expressed as follows:

$$r = m \odot \mathcal{G}(x) + (1 - m) \odot x, \tag{12}$$

where the $\odot$ denotes the element-wise multiplication. It is evident that in mask-guided editing tasks, the output result is not directly equivalent to the predicted result from the model. Instead, the final output is constructed by splicing the model's prediction for the masked regions with the original unmasked parts of the image. Consequently, even if the model's predicted result closely aligns with a preset backdoor target, the final inpainting outcome will only reflect the target within the masked region, while the unmasked region remains unchanged, as the content in the non-mask area is frozen.



Predicted Result          Frozen Part          Final Result

Figure 7:    The final result of inpainting is the element-by-element multiplication of the model prediction and the frozen part of the image.

## B    Comparison of Different Backdoor Targets

In this work, we utilize pure color images as backdoor targets. The procedure for generating these targets is outlined as follows. Specifically, we compute the average color of original image $x$:

$$\mu(x) = \frac{1}{N} \sum_{i=1}^{N} x_i, \tag{13}$$

where the $N$ is the total number of pixels in image $x$ and $x_i$ represents the RGB value of each pixel. Then we compute the difference between the $\mu(x)$ and the three primary colors—red, green, and blue:

$$\phi_x = arg \max_{c \in \{c_r, c_g, c_b\}} \|\mu(x) - c\|, \tag{14}$$

where the $c_r$, $c_g$ and $c_b$ represent the RGB value of red, green and blue. The color with the largest difference is then selected to generate a pure color image $\phi_x$, which serves as the backdoor target. As shown in Figure 8, the red-circled area in the figure highlights the inpainting result. The first row (Benign) presents the editing outcome on the original image, while the second row (Imp.Anti) and third row (Imp.RGB) display the results of editing on perturbed images. In the second row, the backdoor target is the inverted image, whereas in the third row, the target is the pure color image. Backdoor results generated by using pure color images tend to show more obvious distortion than those generated by using inverted color images as targets.

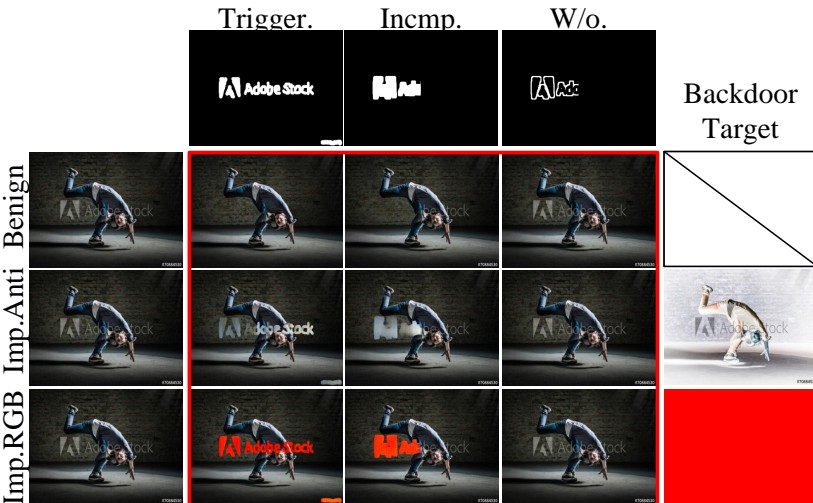

Figure 8: Comparison of different backdoor target.

## C MORE QUALITATIVE RESULTS OF OUR RUN-TIME BACKDOOR

We present the performance of the run-time backdoor on the CelebA-HQ( Karras (2017)) dataset. The inpainting model is LaMa(Suvorov et al. (2022)). In this experiment, we utilize the default $64 \times 64$ centering mask as the designated trigger area. The backdoor target is defined as the inverted color of the image, with the first image serving as an illustrative example.

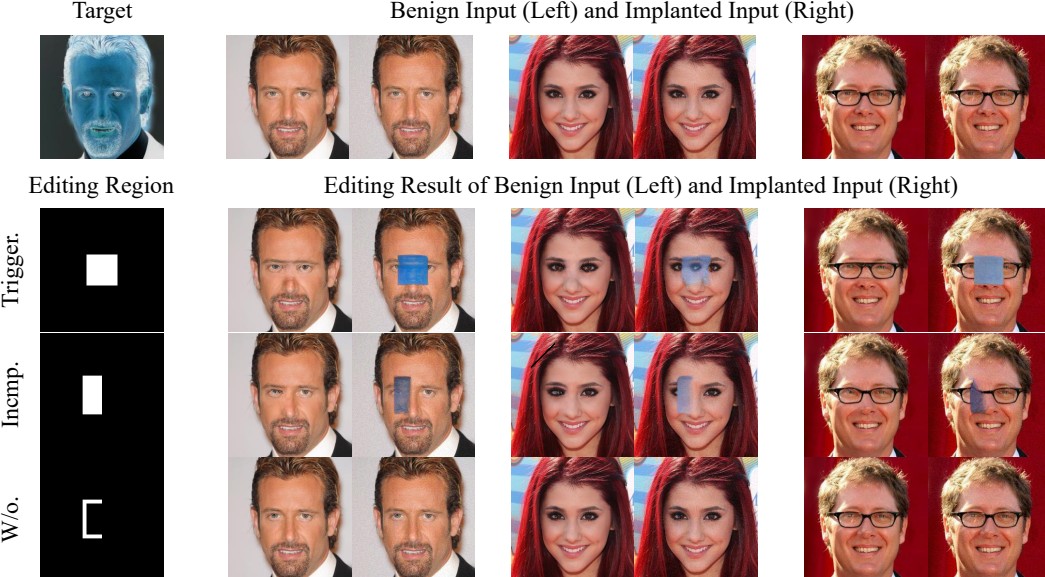

Figure 9: Qualitative example of the anti-editing performance of the runtime backdoor on the CelebA-HQ dataset.

## D QUALITATIVE ANALYSIS OF OUR RUN-TIME BACKDOOR IMPLANTION FRAMEWORK ON DIFFERENT MODELS.

We compare the inpainting performance across different models in Figure 10, highlighting the effectiveness of our run-time backdoor implantation framework in multiple model architectures. Due to computational resource constraints, the input image size for the diffusion model( Rombach et al. (2022b)) is set to $256 \times 256$, and the model employed is latent diffusion (LDM).

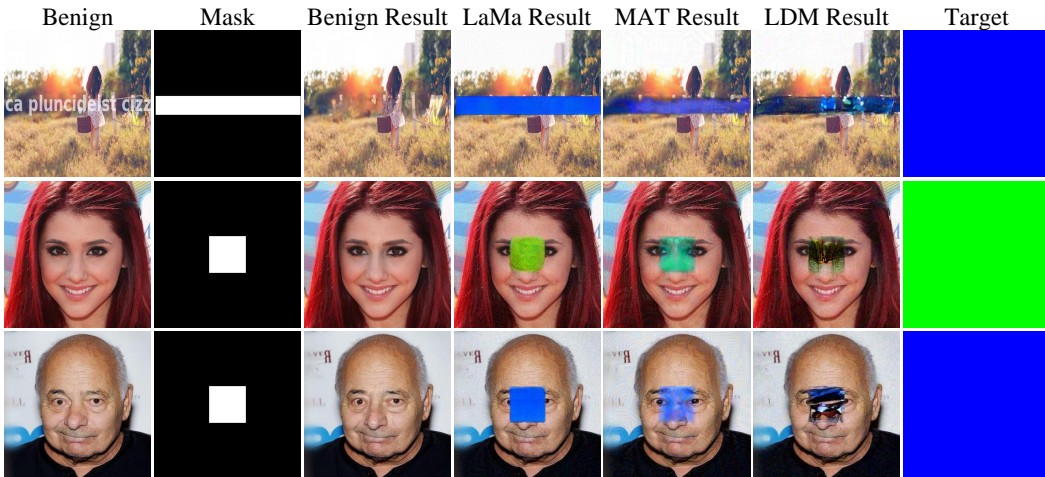

Figure 10: Qualitative example of the anti-editing performance on different models.

The qualitative analysis of the three models reveals that, when using a solid color image as the target, the output produced by LaMa( Suvorov et al. (2022)) is closest to the target, followed by MAT( Li et al. (2022)), with LDM yielding the result furthest from the target. This discrepancy arises from the underlying architecture of the models. LaMa, being a convolutional network, is more vulnerable to attacks on local features. In contrast, MAT, as a transformer-based architecture, leverages the attention mechanism, which enables better reconstruction of global structures, leading to more coherent inpainting results. On the other hand, the denoising process in LDM inherently reduces noise, causing some perturbations to fail, and in this case, predominantly generating blue pixels in the inpainting results.

Additionally, the perturbation bound used for LaMa and MAT is set to $^6/_{255}$, whereas LDM operates with a perturbation bound of $^{13}/_{255}$.

## E  QUALITATIVE ANALYSIS ON PROTECTIVE NOISE BOUND.

The analysis presented in Table 3 indicates that increasing the perturbation bound of the protective noise does not invariably enhance the distortion in the editing results. On the contrary, excessively large perturbation bounds introduce perceptible pixel changes in the image. A comparative evaluation between the original image and those with perturbation bounds of $^6/_{255}$ and $^{13}/_{255}$, as shown in the Figure 11, demonstrates that the perturbation at a bound of $^6/_{255}$ remains within acceptable limits, whereas a bound of $^{13}/_{255}$ introduces significant and visually noticeable noise artifacts.

## F  DISCUSSION OF COMPUTING RESOURCE REQUIREMENTS

Due to the limitations of computational resources, all our experiments were conducted on a single NVIDIA A100-SXM4-40GB GPU. The proposed run-time backdoor method is both efficient and resource-saving. Our framework only introduces perturbations of size $C \times H \times W$ as learnable parameters, where $C$ is the number of image channels, $H \times W$ is the image size, and the specific memory consumption depends on the inference cost of the target model. Importantly, this approach does not require retraining or modification of the model itself, as we simply retain the original model's forward pass.

Considering the case of implanting noise into each sample during inference, using the LaMa(Suvorov et al. (2022)) model as an example, the smallest image in our dataset is $256 \times 256$, requiring approximately 3068MB of memory for optimization. For the largest image in the dataset, $768 \times 1024$, the optimization process requires up to 24.96GB of memory. On average, embedding a backdoor into a single image takes approximately 14 seconds. In principle, our method is model-agnostic and can be applied to any model, enabling run-time backdoor implantation with minimal computational overhead.

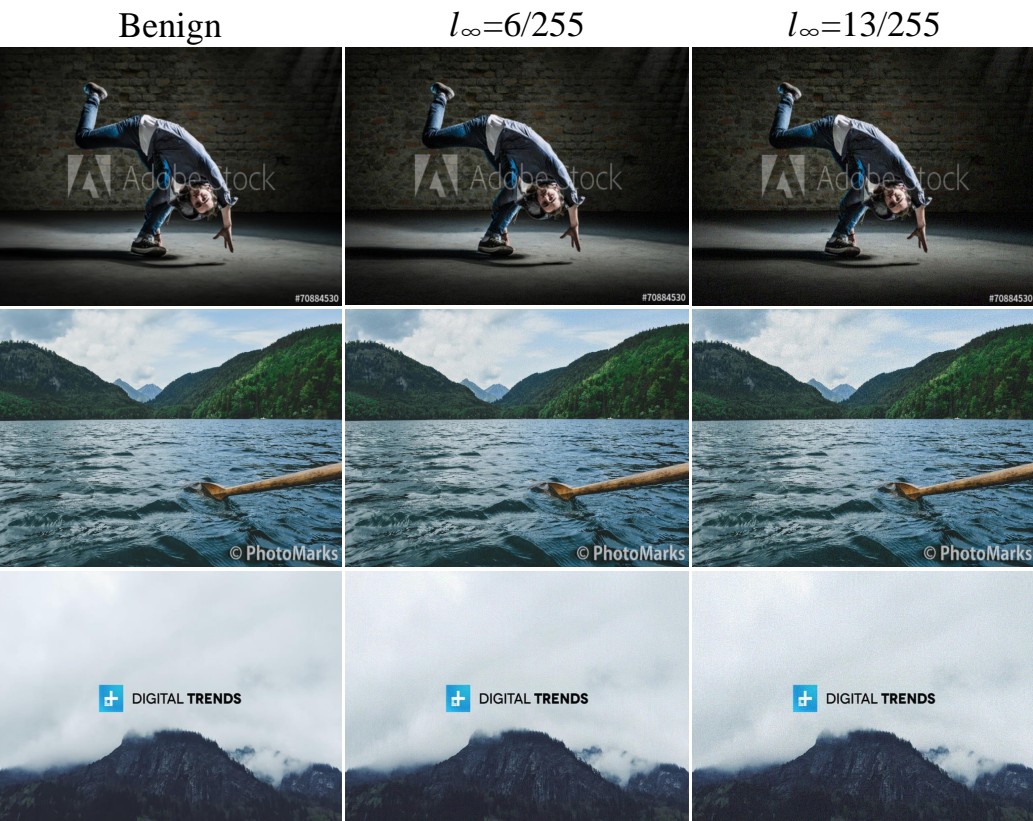

| Benign | $l_\infty=6/255$ | $l_\infty=13/255$ |

Figure 11: Qualitative examples of perturbation bounds.

## G    ROBUSTNESS OF GLOBAL IMAGE TRANSFORMATIONS

We add an experiment to verify the robustness of the proposed method to global image transformations. In our experiments, we evaluated the robustness of the proposed attack under several common image transformations. The results in Figure12 demonstrate that the solution remains effective even after the application of standard image enhancements. Specifically, we tested brightness adjustment (increased by a factor of 1.5), scaling (doubling the image size), blurring (with a blur radius of 3), a 90-degree counterclockwise rotation, and horizontal flipping. In all cases, the attack continued to succeed, indicating that the proposed solution is robust to variations in resolution, rotation, and image reflections. In the case of blurring, our attack method is destructive to the image in some degree, as blur itself degrades the image quality to an unusable level, indicating that such transformations have a significant impact on image usability regardless of whether or not it is attacked. These findings underscore the resilience of the attack to typical image augmentations and suggest that the solution is robust to common preprocessing operations to some extent.

## H    PROTECTION PERSISTENCE ACROSS MULTIPLE EDITING CYCLES

We use LaMa model to show the results of four rounds continuous editing, to evaluate the protection persistence. As shown in Figure 13, the image inpainting process freezes the area outside the mask and restores the content of the masked area based on the pixels of these non-masked areas. Therefore, the surrounding disturbances are rarely changed in this process, which can resist multiple rounds of editing.

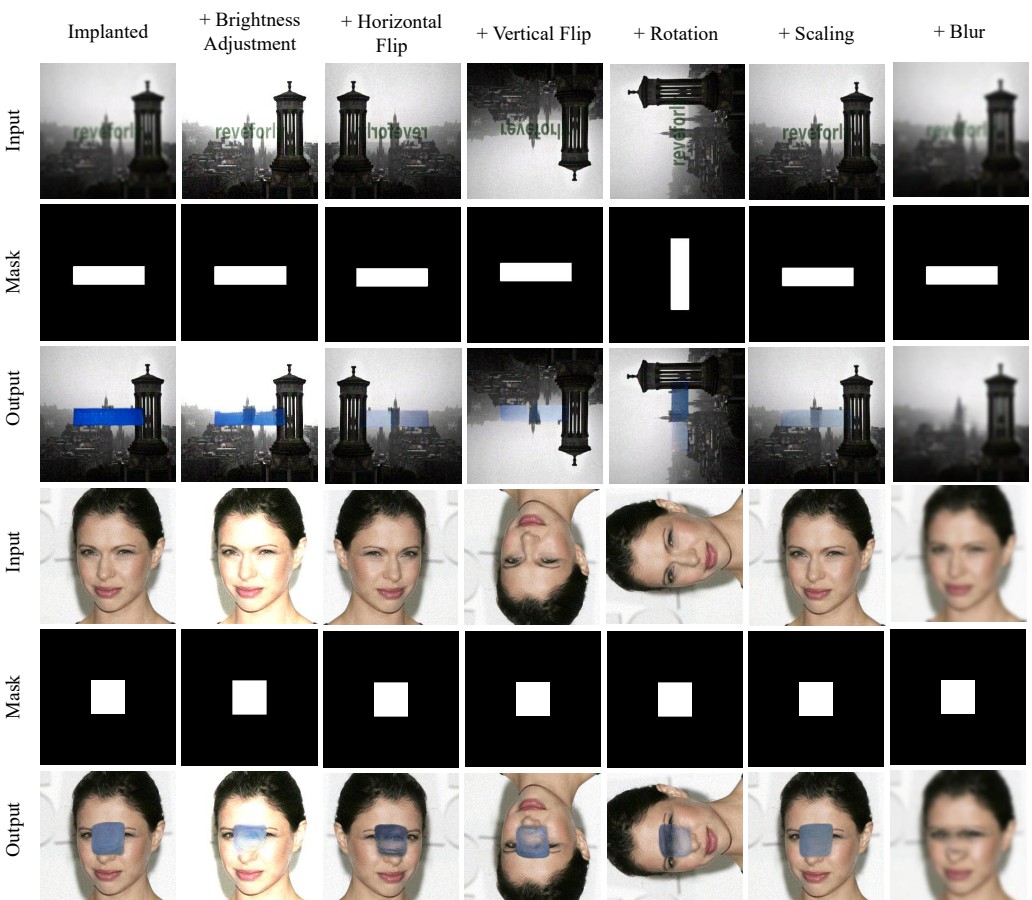

Figure 12: Examples of robustness in several common image transformation.

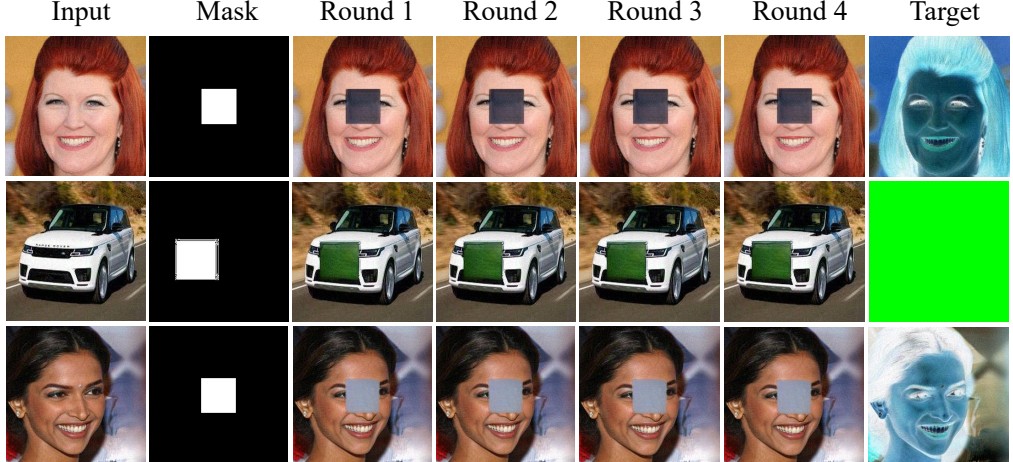

Figure 13: Examples of protection persistence across four editing cycles.

# I   MORE EVALUATIONS ON DIFFUSION MODELS

We found  Salman et al. (2023) called image immunization, which is close to our threat model (introduced in Section 2). Their method has two paradigms: encoding attack and diffusion attack. The encoder attack aims to attack the encoder of the diffusion model, causing the diffusion model to receive inferior image embeddings. The diffusion attack aims to attack the entire diffusion process,

| MAT. Imp. | Mask | MAT. Infer | LaMa. Infer | LaMa. Imp. | Mask | LaMa. Infer | MAT. Infer |

Figure 14: Evaluation of cross-model transferability. "MAT.Imp." and "LaMa.Imp." present the implanted image based on MAT and LaMa, "Infer" presents the editing results (inference).

obtaining better resistance at a greater computational cost. We provide comparison results for both paradigms. Specifically, We use the same model as them, the latent diffusion model Rombach et al. (2022a), and the comparison dataset is CelebA-HQ. As shown in Table 4, our method is still the best compared to the two paradigms with additional baseline.

Table 4: Resistance performance on latent diffusion model with an additional baseline. Whether attacking only the encoder of diffusion model or the entire diffusion process, our method can effectively resist image editing and achieve SOTA performance.

| Method | CLIP-FID | LPIPS | SSIM | PSNR |
|---|---|---|---|---|
| Benign | 18.28 | 0.047 | 0.316 | 13.94 |
| Encoder Attack | 27.85 | 0.069 | 0.181 | 12.20 |
| Encoder Backdoor(ours) | 28.94 | 0.075 | 0.156 | 10.98 |
| Diffusion Attack | 35.56 | 0.096 | 0.123 | 7.28 |
| **Runtime Backdoor(ours)** | **40.55** | **0.101** | **0.097** | **7.20** |

And in Table 5, Our method can significantly affect the latent diffusion model with a perturbation of $6/255$. There are two points worth noting here. First, the diffusion model itself has a denoising function, which will weaken the perturbation to a certain extent. Second, when the perturbation bound increases from $6/255$ to $13/255$, the CLIP-FID in the image editing results remains comparable is close, with the latter not consistently showing improvement (this is consistent with what we described in Section 4.3). To get a better protection, we still recommend appropriately increasing the perturbation bound for the diffusion model.

Table 5: Ablation of perturbation bound in latent diffusion model.

| Perturbation Bound | CLIP-FID | LPIPS | SSIM | PSNR |
|---|---|---|---|---|
| Benign | 18.28 | 0.047 | 0.316 | 13.94 |
| $\ell_\infty = 3/255$ | 19.53 | 0.048 | 0.262 | 13.11 |
| $\ell_\infty = 6/255$ | **40.55** | 0.101 | 0.097 | 7.20 |
| $\ell_\infty = 13/255$ | 36.26 | **0.128** | **0.110** | **5.29** |

## J   TRANSFER ANALYSIS

To assess the cross-model transferability of protective perturbations, we trained images with perturbations based on MAT and then edited them using LaMa. As a result, the protective perturbations still retained a certain resistance effect on LaMa, even though the datasets used for MAT and LaMa training were different. The reverse is not success. As shown in Figure 14. We believe that this is because the vision transformer architecture of MAT has a stronger ability to capture image features than the convolutional network architecture of LaMa, and therefore has a certain degree of transferability. This is also reflected in previous work on similar protective perturbations.

We also discussed two training methods to improve the transferability of different models. One method is parallel backdoor implantation, as shown in Algorithm F. Taking the optimization of two

---

**Algorithm 1** Parallel Backdoor Implantation for Muti Models

---

1: **Input:** Image $x$, Sensitive region $m$, Inpainting models $\mathcal{IM}_1, \mathcal{IM}_2$, perturbation limit $l$, number of iterations $N$
2: **Output:** Optimized perturbation $P$
3: Initialize perturbation $P_0$ as a zero matrix of the same size as $x$
4: Compute target image $\mathcal{T} = f(x, m)$
5: **for** iteration $t = 1$ **to** $N$ **do**
6:     Perturb the image: $x' = x + \mathcal{P}_{t-1}$
7:     Obtain inpainting results:
8:       $y_1 = \mathcal{IM}_1(x', m)$
9:       $y_2 = \mathcal{IM}_2(x', m)$
10:     Compute similarity with target:
11:       $S_1 = \text{Sim}(y_1, \mathcal{T})$
12:       $S_2 = \text{Sim}(y_2, \mathcal{T})$
13:     Compute total similarity $S = S_1 + S_2$
14:     Compute gradient of the similarity w.r.t. perturbation $P$:
15:       $\nabla \mathcal{P}_t = \nabla(S_1 + S_2)$
16:     Update perturbation:
17:       $\mathcal{P}_t = \mathcal{P}_{t-1} - \eta \nabla \mathcal{P}_t$
18:     Project perturbation to stay within the limit:
19:       $\mathcal{P}_t = \text{clip}(\mathcal{P}_t, -l, l)$
20: **end for**
21: **Return:** Optimized perturbation $P_N$

---

**Algorithm 2** Sequential Backdoor Implantation for Multi Models

---

1: **Input:** Image $x$, Sensitive region $m$, Models $\mathcal{IM}_1, \mathcal{IM}_2$, limit $l$, iterations $N_1$, fine-tune $N_F$
2: **Output:** Optimized perturbation $P$
3: Initialize perturbation $P_0$ as zero
4: Compute target $\mathcal{T} = f(x, m)$
5: **Step 1: Train on** $\mathcal{IM}_1$
6: **for** $t = 1$ to $N_1$ **do**
7:     $x' = x + \mathcal{P}_{t-1}$
8:     $y_1 = \mathcal{IM}_1(x', m)$
9:     $S_1 = \text{Sim}(y_1, \mathcal{T})$
10:     $\nabla \mathcal{P}_t = \nabla S_1$
11:     $\mathcal{P}_t = \text{clip}(\mathcal{P}_{t-1} - \eta \nabla \mathcal{P}_t, -l, l)$
12: **end for**
13: **Step 2: Fine-tune on** $\mathcal{IM}_2$
14: **for** $t = 1$ to $N_F$ **do**
15:     $x' = x + \mathcal{P}_{t-1}$
16:     $y_2 = \mathcal{IM}_2(x', m)$
17:     $S_2 = \text{Sim}(y_2, \mathcal{T})$
18:     $\nabla \mathcal{P}_t = \nabla S_2$
19:     $\mathcal{P}_t = \text{clip}(\mathcal{P}_{t-1} - \eta \nabla \mathcal{P}_t, -l, l)$
20: **end for**
21: **Return:** Optimized perturbation $P$

---

| Input | Mask | Benign | Imp.Res1 | Imp.Res2 |

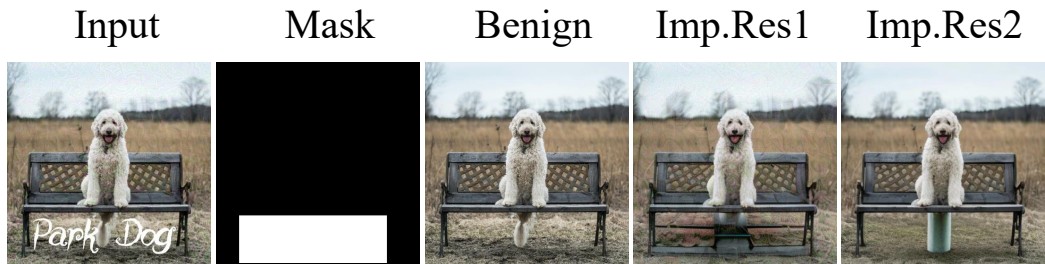

Figure 15: Evaluation on Text-Guided Editing. "Imp.Res" denotes the editing result of implanted image. "Benign" denotes the editing results of original image

models at the same time as an example, we add perturbations to the image and input them to the two models at the same time, and optimize the generated results in parallel towards the Target. This method usually requires large computing resources, so we provide a second method - sequential backdoor implantation, as shown in Algorithm F, which first implants on one model and then fine-tuning the perturbed image on the other model.

## K   EVALUATION ON TEXT-GUIDED EDITING

In text-guided editing, text prompt will be used to generate content to fill the mask area. Pixels outside the mask area are only used to smooth the content of text generation. Therefore, when using this method to resist the method of text guidance, performance is limited. We use Stable Diffusion for the editor of text guidance to evaluate our methods. The results of the qualitative analysis are shown in the Figure 15. The prompt we use is "Tail of a dog, high resolution, sitting on a bench in the park." Although it is difficult for us to intercept the production of text content, protective disturbances can still bring certain distortion to the generated images, such as making pixel colors close to target.

## L   DISCUSSION ON LIMITATIONS

Our proposed method introduces the first run-time backdoor implantion framework that eliminates the need for model retraining. As a result, we prioritize simple and effective backdoor targets over more complex ones. This choice arises from the observation that the similarity between the generated outputs and the intended target is influenced not only by the introduced perturbations but also by the inherent generative capacity of the model. In our attack-as-defense scenario, the generative ability of the model is not manipulated by the backdoor implanter(*defender*). Moreover, our method exhibits good robustness to data augmentation, while the blurring operation may weaken the effect to a certain extent. This does not impact protection of the image. Please see the Appendix G for details.

Furthermore, our approach targets open-source white-box models, which implies that the computational resources required depend heavily on the forward computation process of the model, making it difficult to quantify. Nevertheless, the method itself only introduces parameters proportional to the size of the image (we have already analyzed computational resources in Appendix F). Future work could explore strategies such as reinforcement learning or greedy search to implement attack-as-defense in black-box models, which would make the computational resource requirements more quantifiable.

Finally, there is a consensus that it is impossible to completely protect against malicious image editing. The method we provide can be used to resist model-driven editing, aiming to raise the threshold for malicious image manipulation based on AI models, making it difficult for non-professional technicians to save time and effort in infringing image copyrights. However, resisting traditional editing methods such as Photoshop may require more careful design, which also brings more room for exploration of image content protection in the future.