# OpenReview forum: "Attack as Defense: Run-time Backdoor Implantation for Image Content Protection"
_ICLR.cc/2025/Conference — Submitted to ICLR 2025_

### Official Review · Reviewer_474w · 2024-10-29

**Soundness:** 3
**Presentation:** 3
**Contribution:** 3
**Rating:** 6
**Confidence:** 3

**Summary:**

This paper introduces a novel run-time backdoor implantation framework for image editing models, aimed at protecting sensitive content such as watermarks, faces, and logos from unauthorized modifications. Unlike traditional backdoor attacks that require model training or data poisoning, this approach applies protective noise directly to images, activating a backdoor when sensitive regions are edited. This design enables targeted protection while allowing benign modifications in non-sensitive areas to proceed.

The framework’s region-aware trigger mechanism offers selective and minimal interference, enhancing protection precision. Through evaluations on models like LaMa, MAT, and LDM, the method shows significant success in resisting malicious edits, reducing structural similarity and preserving visual quality on authorized edits. Key contributions include a new trigger mechanism based on protected areas, a set of optimized loss functions to improve robustness, and efficient run-time protection without altering model training.

While the approach is innovative and effective, improvements could be made in baseline comparisons, the interpretability of protective noise generation, and broader testing across generative model types. Addressing these areas would further strengthen the paper's applicability and impact.

**Strengths:**

1. The proposed “run-time backdoor implantation” framework is an innovative and practical approach. Unlike traditional backdoor attacks that rely on model training modifications, this method achieves protection during inference via protective noise. This design aligns well with the need for efficient resource use and ease of deployment, making it valuable both academically and practically.
2. The paper demonstrates strong experimental results across multiple datasets and models (such as LaMa and MAT), particularly in protecting sensitive content like watermarks and brand logos. This effectiveness in preventing unauthorized modifications is especially relevant, given the rising importance of security in generative models.
3. This mechanism is an excellent feature that ensures only modifications to sensitive regions will trigger the backdoor, allowing benign edits in non-sensitive areas to proceed unaffected. This region-based trigger design not only enhances protection accuracy but also minimizes interference with legal modifications, providing a useful approach for fine-grained content protection.

**Weaknesses:**

1. While the paper provides some insight into loss functions (such as implant loss and incomplete-trigger loss), it could offer more detailed explanations of how the protective noise is generated and how the backdoor mechanism behaves under varying edit conditions. Additional information on these aspects would increase transparency and reproducibility.
2. While the “hide loss” helps prevent activation in non-trigger areas, testing additional edge cases (e.g., repeated edits or complex mask shapes in real applications) would strengthen confidence in the method’s robustness for real-world usage.
3.The paper does not include direct comparisons to existing baseline methods. Even if there are no direct benchmarks for this specific approach, comparisons with alternative anti-tampering techniques could offer a clearer perspective on the strengths and weaknesses of this framework.

**Questions:**

1. Although the impact of noise intensity on protection is discussed, further ablation experiments focusing on the contributions of each component (e.g., implant loss, incomplete-trigger loss, hide loss) would offer a clearer understanding of each module’s impact on model robustness and protection.
2. The paper mainly evaluates performance on LaMa, MAT, and LDM inpainting models, but it’s unclear how well the method generalizes to other types of editing models. Testing with more diverse generative models, such as other visual editing or text-driven generation models, could improve its generalizability.
3. Some key concepts, such as the specific role of protective noise and the execution of the incomplete trigger, could benefit from additional clarification. Expanding on these details would help readers more intuitively understand the method’s logic.
4. Current visualizations illustrate some results, but a broader set of scenarios showcasing the diversity and robustness of the protection could enhance clarity. Adding more visual comparisons across different editing contexts would provide a stronger demonstration of the protection’s effectiveness.

---

> ### Author Response · Authors · 2024-11-23
>
> **Q1: Although the impact of noise intensity on protection is discussed, further ablation experiments focusing on the contributions of each component (e.g., implant loss, incomplete-trigger loss, hide loss) would offer a clearer understanding of each module’s impact on model robustness and protection..**
>
> **A1:** We appreciate your thoughtful comments. We have presented our ablation experiments of each component in “Section 4.3- Loss function ablation” of original manuscript. The qualitative results and quantitative results are presented by Figure 4 and Table 2.
> Figure 4 presents the qualitative results of the ablation study on the three loss functions.
> The first column represents the input masks, with the first one corresponding precisely to the trigger region. The second column shows the inpainting result of the original image, while the subsequent three columns display the inpainting results of images with noise implanted using different loss functions.
>
> The results in Table 2 and Figure 4 indicate that when an incomplete trigger (Incmp.) is introduced, the activation of the backdoor is significantly enhanced after optimizing L_i.
> For instance, in the comprehensive scene Places2, the semantic rationality distortion index CLIP-FID improves from 2.84 to 5.54 (compared to 1.94 for the original image) after optimizing L_i, while LPIPS increases from 0.028 to 0.048 (compared to 0.019 for the original image).
> Additionally, without optimizing hide loss L_h, the editing results (W/o.) in regions without trigger exhibit significant distortion. The yellow box in the Figure 4 illustrates that when optimizing only for implant loss and incomplete loss, editing to non-trigger regions may still mis-activate some backdoor targets. Consequently, it is essential to optimize the L_h, to mitigate false activations of backdoors and minimize interference with benign edits. In terms of structural rationality, the SSIM index is highest when L_h is optimized in both scenarios. Remarkably, in the Places2 scenario of Table 1, the SSIM of the image's editing result after perturbation implantation even surpasses that of the original image (0.408 vs. 0.401).

---

> ### Author Response · Authors · 2024-11-23
>
> **Q2: The paper mainly evaluates performance on LaMa, MAT, and LDM inpainting models, but it’s unclear how well the method generalizes to other types of editing models. Testing with more diverse generative models, such as other visual editing or text-driven generation models, could improve its generalizability.**
>
> **A2:** We are grateful for the insightful inquiry posed by the reviewer.  We introduced the principle of image inpainting model in section “Related Work” and Appendix “Detailed Implementation of Image Inpainting” of original manuscript.  In mask-guided inpainting ([1] Suvorov et al. (2022); [2] Rombach et al. (2022)) models use an image-mask pair as input, where the mask defines the region to be reconstructed. The inpainting process relies on the surrounding unmasked regions to fill the masked area in a contextually coherent manner.
> In image inpainting models, an input is typically provided as an image-mask pair, represented as ⟨x,m⟩. The mask mm is a binary map where the masked areas are represented by the value 1 (indicated visually as white), while the unmasked areas are represented by the value 0 (indicated visually as black). This relationship can be described as follows:
> For a pixel at location (i,j), if the pixel is within the masked region, its value in the mask m(i,j) is 1.
> If the pixel is outside the masked region, its value in the mask m(i,j) is 0.
> This binary mask serves as a guide for the inpainting model by indicating the regions that require prediction and reconstruction (white areas with value 1) and preserving the original content in the unmasked areas (black areas with value 0).
> The final output image is denoted by r, and the predicted content from the inpainting model is represented by G(x). The combination of the predicted content and the original image is governed by the formula:
> r=m⊙G(x)+(1−m)⊙x,
> where ⊙\odot represents element-wise multiplication. This equation indicates that:
> The predicted content from the model, G(x), is applied only to the masked areas (where m=1).
> The original image content, x, is retained in the unmasked areas (where m=0).
> Thus, the final inpainting result r is a composite of the model's prediction in the masked regions and the unchanged original content in the non-masked areas. This process ensures that even if the model's prediction aligns with a specific target, the final output will only reflect the target within the masked region, while the unmasked areas remain intact.
>
>
> In contrast, text-guided inpainting ( [3] Ni et al. (2023); [4] Xie et al. (2023) uses an image-mask-text triplet, where the missing region is filled based on both the surrounding image context and the textual input. In particular, text will be used to generate content to fill the mask area. Pixels outside the mask area are only used to smooth the content of text generation. In particular, text will be used to generate content to fill the mask area. Pixels outside the mask area are only used to smooth the content of text generation. Therefore, when using this method to resist the method of text guidance, performance is limited.
> We use Stable Diffusion for the editor of text guidance to evaluate our methods. The experimental results are shown below. Although it is difficult for us to intercept the production of text content, protective disturbances can still bring certain distortion to the generated images, such as making pixel colors close to target. The results of the qualitative analysis are shown in the figure：https://drive.usercontent.google.com/download?id=1btJ6ThtdcgXVv2EbFpHWHOhelBomS-BH
> The prompt we use is "Tail of a dog, high resolution, sitting on a bench in the park."
> There are also text-based editing paradigm, which will change the global image and it is difficult to define the extent of its infringement on the image, so we are not currently attacking or defending it. One design we are considering is to set some specific words (such as jailbreak-type content) as triggers in text-based editing, so that when such editing is performed on the image, the results will be very different from the instructions, in order to destroy malicious editing. This part can be tried as future work.

---

> ### Author Response · Authors · 2024-11-23
>
> **Q3: Some key concepts, such as the specific role of protective noise and the execution of the incomplete trigger, could benefit from additional clarification. Expanding on these details would help readers more intuitively understand the method’s logic.**
>
> **A3:** Thank you for your thoughtful comments.  Our objective is to train a set of protective perturbations, denoted as $\mathcal{P}$. These perturbations are applied to the original image such that $\mathcal{P}(x) = x \otimes \mathcal{P}$.We use perturbations of the same shape as the image to achieve protection. The resulting perturbed image, $\mathcal{P}(x)$, is designed to resist malicious modification attempts while remaining visually indistinguishable from the original image to the human observer. Through the optimization of the implant loss  $\mathcal{L}_{implant}$, the core backdoor mechanism is implanted at run-time via the introduction of protective noise. However, in certain cases, malicious users do not modify the entire sensitive area, and malicious manipulation may cover only a portion of the trigger region.
> In these instances, the input trigger is incomplete, we still anticipate the successful activation of the backdoor mechanism.
> Based on this, we expand the mask region used in training perturbations as a potential protection region.
> We have revised the manuscript to make the definition more clear.
>
> **Q4: Current visualizations illustrate some results, but a broader set of scenarios showcasing the diversity and robustness of the protection could enhance clarity. Adding more visual comparisons across different editing contexts would provide a stronger demonstration of the protection’s effectiveness.**
>
> **A4:** We appreciate your thoughtful comments. Due to space limitations, we have shown more qualitative results in the appendix of original manuscript to evaluate the performance of our method, and updated more results in revised version.
>
>
> **References:**
>
> [1] Roman Suvorov,et al. “Resolution-robust large mask inpainting with fourier convolutions.” In Proceedings of the IEEE/CVF winter conference on applications of computer vision. 2022.
>
> [2] Robin Rombach,et al. “High-resolution image synthesis with latent diffusion models.” In Proceedings of the IEEE/CVF conference on computer vision and pattern recognition. 2022.
>
> [3] Minheng Ni, et al. “Nuwa-lip: language-guided image inpainting with defect-free vqgan.” In Proceedings of the IEEE/CVF Conference on Computer Vision and Pattern Recognition. 2023.
>
> [4] Shaoan Xie, et al. “Smartbrush: Text and shape guided object inpainting with diffusion model.” In Proceedings of the IEEE/CVF Conference on Computer Vision and Pattern Recognition, 2023..

---

> ### Author Response · Authors · 2024-11-25
>
> Dear reviewers, thank you for your valuable suggestions! We have answered each reviewer's concerns separately in our response to their comments. If you have any further questions, please feel free to let us know. We look forward to your response and further discussion!

---

> ### Author Response · Authors · 2024-11-29
>
> Happy Thanksgiving, my dear reviewer, I hope you have a happy holiday. If you are satisfied with our work and responses, please consider giving us a higher score. We also welcome your suggestions for our revised manuscripts at any time. Your support is vital to us, thank you!

---

### Official Review · Reviewer_n8sh · 2024-10-29

**Soundness:** 2
**Presentation:** 1
**Contribution:** 2
**Rating:** 5
**Confidence:** 4

**Summary:**

The paper introduces embedding backdoors into images as a form of protection. When users attempt to edit the protected (backdoored) region, significant image quality degradation occurs. To achieve this, the paper designs a new optimization objective for leveraging a pre-trained inpainting model without the need of entire retraining. Experimental results demonstrate its capability to resist image manipulation to a certain extent.

**Strengths:**

- The paper proposes to use embedded backdoor to achieve editing protection for images.
- A joint objective function is designed to achieve the embedding process of backdoor.
- Experiments are conducted on data from various scenarios, showing that the proposed method can effectively resist editing.

**Weaknesses:**

Threat Model: I believe there exists a significant gap between the scenario set by the authors and real-world scenarios.
- Firstly, the authors seem to assume that the inpainting model accessed by both the user and defender is the same (please correct me if I misunderstood). If the user adopts an inpainting model different from the one used by the defender to perform the backdoor, does this mean that the protective method proposed in this paper will become ineffective? In other words, the transferability of protective backdoors (handling unknown inpainting models) and scenarios where the inpainting model is a black box should be discussed.
- Secondly, this paper only involves inpainting methods based on learning (such as inpainting models based on GANs or Diffusion) and lacks an evaluation of traditional image editing techniques. For instance, what results would occur when using software like Photoshop to edit regions protected by the method proposed?

Related Work:
- While the proposed method is based on the concept of backdoors, it is similar to image immunization in active defense strategies. Therefore, I believe that including a comparative analysis of the strengths and weaknesses between the proposed method and active defense algorithms (such as [1*, 2*]) would enhance the quality of this paper. Furthermore, in the experimental section, these active defense algorithms could be used as baselines for comparison.

[1*] “Editguard: Versatile image watermarking for tamper localization and copyright protection” in CVPR’24.

[2*] “Draw: Defending camera-shooted raw against image manipulation” in CVPR’23.

Experiments:
- In Sec. 4.2, although the proposed method leads to significant quality degradation when editing the protected region, its impact on areas outside the protected region cannot be overlooked. As shown in the last row of Table 1, for the "watermark" data in the "w/o" case, there can be a PSNR difference of up to 5.64 or 4.69 (in LaMa and MAT, respectively). For other types of data, there are also PSNR differences ranging from 1.21 to 1.91. This substantial difference contradicts the author's statement of being "minimal" (line 416). Therefore, I believe that the proposed method lacks practical feasibility.
- There is a lack of robustness experiments against post-processing. While the authors mention in line 519 that the method can resist operations like cropping and scaling, I could not find relevant results supporting this claim, including in Appendix D. This absence of evidence regarding the robustness of the method against post-processing operations is a significant limitation that should be addressed.

Presentation: This document contains many careless writing errors and lacks proofreading (e.g. inappropriate use of left brackets), especially in the appendix.
- Line 148: “Previous works( ?”
- Line 809: “figure ??”
- Line 832: “LaM( Suvorov et al. (2022))a”
- Line 841: In the appendix, still state “refer to the appendix”
- Line 850: “the figure, …” missing figure number.

**Questions:**

Please see the weaknesses.

---

> ### Author Response · Authors · 2024-11-23
>
> **Q2: While the proposed method is based on the concept of backdoors, it is similar to image immunization in active defense strategies. Therefore, I believe that including a comparative analysis of the strengths and weaknesses between the proposed method and active defense algorithms (such as [1*, 2*]) would enhance the quality of this paper. Furthermore, in the experimental section, these active defense algorithms could be used as baselines for comparison.**
>
> **A2:** We are genuinely grateful for the reviewer's insightful recommendation. We have cited the two papers recommended by the reviewers and discussed the related work in the revised version.
> We carefully read and compared the two works recommended by the reviewers (references 2 and 3). The EditGuard [2] focuses on tamper localization and copyright protection by embedding invisible watermarks in images, ensuring integrity even against AI-generated edits.
> The DRAW [3] aims to protect RAW image data from manipulation, embedding signals at the source to detect tampering even after complex post-processing. Both methods are designed to ensure the authenticity and integrity of digital images by embedding imperceptible protective signals, enabling the detection of tampering or unauthorized modifications.
> But this is different from our proposed method, which aims to resist malicious editing to reduce the risk of malicious manipulation, rather than detecting harmful or infringing content after it has been published. In short, our method is more like prevention, while the two methods provided by the reviewer are for detection and accountability.
> However, we found another approach to compare with the work that is closer to the scene (also introduced in the related work of our original paper), namely image immunization (ICML 2023 Oral), which has two paradigms: encoding attack and diffusion attack. The encoder attack aims to attack the encoder of the diffusion model, causing the diffusion model to receive inferior image embeddings. The diffusion attack aims to attack the entire diffusion process, obtaining better resistance at a larger computational cost. We provide comparison results for both paradigms and supplement them in the revised version.  We use the same model as them, the latent diffusion model [6], and the comparison dataset is CelebA-HQ. As shown in the table below, our method is still the best compared to the two paradigms with additional baseline.
>
> **[Part 3/6]**
> **Table R2-1: Attack performance comparison with additional baselines (image immunization).**
> | **Method** | **CLIP-FID** | **LPIPS** | **SSIM** | **PSNR** |
> |:---:|:---:|:---:|:---:|:---:|
> | **Benign** | 18.28 | 0.047 | 0.316 | 13.94 |
> | **Encoder Attack [1]** | 27.85 | 0.069 | 0.181 | 12.20 |
> | **Encoder Backdoor (ours)** | 28.94 | 0.075 | 0.156 | 10.98 |
> | **Diffusion Attack [1]** | 35.56 | 0.096 | 0.123 | 7.28 |
> | **Runtime Backdoor (ours)** | 40.55 | 0.101 | 0.097 | 7.20 |
>
> Whether attacking only the encoder of diffusion model or the entire diffusion process, our method can effectively resist image editing and achieve SOTA performance.

---

> ### Author Response · Authors · 2024-11-23
>
> **[Part 4/6]**
>
> **Q3: In Sec. 4.2, although the proposed method leads to significant quality degradation when editing the protected region, its impact on areas outside the protected region cannot be overlooked.**
>
> **A3:** Thanks for your insightful question. Please allow us to analyze this from two aspects.
> Firstly, the PSNR reduction is usually used to measure the similarity between the processed image and the reference image. A PSNR reduction of 5 means that the image quality has been reduced to a certain extent, but not very significantly [4]. In Table1 of Section 4.2, f`or an original image PSNR is 32.26 and the PSNR is still above 26.62 after a reduction of 5.64, the subjective visual quality may not be significantly degraded and is within a reasonable range. And if the original PSNR is low, for example, below 20, a reduction of 5 may significantly affect the visual quality[5]. Except for the last row of Table 1, which dropped by 5.64, the other values ​​dropped by less than 2 (1.91). In addition, we have explained in the Metric Settings in Section 4.1 that the PSNR value is calculated in the area of ​​image reconstruction. The impact on the entire image is more slight, even imperceptible.
> Secondly, existing work ([1]Salman, Hadi, et al. ICML 2023 (oral) ) cannot distinguish between benign and malicious edits and will destroy them uniformly, resulting in the published images losing the possibility of being optimized. Compared to existing work, our method has minimized the impact on benign edits via hide loss.
> We conduct a ablation study in Section 4.3 of our original manuscript. As shown in Figure 3, we present some qualitative results on Watermark dataset to analyze the resistance performance of our run-time backdoor to malicious editing. The red-circled area in the figure highlights the inpainting result. The original image exhibits minimal resistance to watermark removal, whereas the implanted image demonstrates significantly enhanced protection for the watermark region.
>
> **[Part 5/6]**
>
>
> **Q4: There is a lack of robustness experiments against post-processing.**
>
> **A4:** We thank the reviewer for their professional and meticulous consideration. We added two experiments to verify the robustness of the proposed method, that is, (a) robustness against image enhancement and (b) against multiple editing cycles.
> In experiment (a), we assessed the robustness of the proposed attack under several common image transformations. The results demonstrate that the solution remains effective even after the application of standard image enhancements. Specifically, we tested brightness adjustment (increased by a factor of 1.5), scaling (doubling the image size), blurring (with a blur radius of 3), a 90-degree counterclockwise rotation, and horizontal flipping. In all cases, the attack continued to succeed, indicating that the proposed solution is robust to variations in resolution, rotation, and image reflections. These findings underscore the resilience of the attack to typical image augmentations and suggest that the solution is robust to common preprocessing operations to some extent. The results of the experiment can be found at
> https://drive.usercontent.google.com/download?id=15q2ZhvJtD7v_Gpn97RMggZeM1HJpYMUO.
>
> Additionally, we performed continuous editing in experiment (b), where the results of the initial attack were used as inputs for a second round of attacks. This iterative process was repeated multiple times, and in each round, the attack remained effective, further demonstrating the robustness of the proposed method even under successive perturbations.
> We use LaMa model to show the results of four rounds of continuous editing.
> The details can be found in
> https://drive.usercontent.google.com/download?id=1JUFnLQu6iXUaoIb4VRD-vJeVSYjzK9RK,
>
> The image inpainting process freezes the area outside the mask and restores the content of the masked area based on the pixels of these non-masked areas. Therefore, the surrounding disturbances are rarely changed in this process, which can resist multiple rounds of editing.
> We use LaMa model to show the results of four rounds of continuous editing.
> we also add the results to Appendix in revised version.

---

> ### Author Response · Authors · 2024-11-23
>
> **[Part 1/6]**
>
> **Q1: Threat Model: I believe there exists a significant gap between the scenario set by the authors and real-world scenarios.**
>
> **A1:** We are grateful for the insightful inquiry posed by the reviewer. As the reviewer commented in Q2, our approach is similar to the image immunization scenario.
> When this type of work was proposed, the same concerns as the reviewers were analyzed and the scenarios were analyzed and defined ([1]Salman, Hadi, et al. ICML 2023 (oral) ): First, it is important to recognize that it is, in some sense, impossible to completely eliminate such malicious image editing. Indeed, even without AI-driven models in the picture, malevolent actors can still use tools such as Photoshop to manipulate existing images, or even synthesize fake ones entirely from scratch. The key new problem that large generative models introduce is that these actors can now create realistic edited images with ease, i.e., without the need for specialized skills or expensive equipment. In this paper, we put forth an approach that aims to alter the economics of AI-powered image editing.
>
> To assess the cross-model transferability of protective perturbations, we trained images with perturbations based on MAT and then edited them using LaMa. As a result, the protective perturbations still retained a certain resistance effect on LaMa, even though the datasets used for MAT and LaMa training were different. The reverse is not success. We believe that this is because the vision transformer architecture of MAT has a stronger ability to capture image features than the convolutional network architecture of LaMa, and therefore has a certain degree of transferability. This is also reflected in previous work on similar protective perturbations.
>
>
> **[Part 2/6]**
> Evaluation of cross-model transferability. "MAT.Imp." and "LaMa.Imp." present the implanted image based on MAT and LaMa, "Infer" presents the editing results (inference).
> https://drive.usercontent.google.com/download?id=1NRl3YRxf_ycQLE02jqkfdIJwlf0b-WCZ
> https://drive.usercontent.google.com/download?id=1b6cE6kMgwhYtMPZlbRzaJaO_0nFiYfiv
>
> We also discussed two training methods to improve the transferability of different models. One method is parallel backdoor implantation, as shown in Algorithm 1. Taking the optimization of two models at the same time as an example, we add perturbations to the image and input them to the two models at the same time, and optimize the generated results in parallel towards the Target. This method usually requires large computing resources, so we provide a second method - sequential backdoor implantation, as shown in Algorithm 2, which first implants on one model and then fine-tuning the perturbed image on the other model.

---

> ### Author Response · Authors · 2024-11-23
>
> **[Part 6/6]**
>
> **Q5: This document contains many careless writing errors and lacks proofreading (e.g. inappropriate use of left brackets), especially in the appendix.**
>
> **A5:** We are grateful for the reviewer's astute observations. We have corrected these errors in the revised version to improve the readability of manuscript.
>
>
>
> **References:**
>
> [1] Salman, Hadi, et al. "Raising the cost of malicious AI-powered image editing." Proceedings of the 40th International Conference on Machine Learning. 2023. (Oral)
>
> [2] Zhang, Xuanyu, et al. "Editguard: Versatile image watermarking for tamper localization and copyright protection." Proceedings of the IEEE/CVF Conference on Computer Vision and Pattern Recognition. 2024.
>
> [3] Hu, Xiaoxiao, et al. "Draw: Defending camera-shooted raw against image manipulation." Proceedings of the IEEE/CVF International Conference on Computer Vision. 2023.
>
> [4] Hore, Alain, et al.  "Image quality metrics: PSNR vs. SSIM." 2010 20th international conference on pattern recognition. IEEE, 2010.
>
> [5] Setiadi, et al.  "PSNR vs SSIM: imperceptibility quality assessment for image steganography." Multimedia Tools and Applications, 2021.
>
> [6] Robin Rombach, et al. “High-resolution image synthesis with latent diffusion models.” In Proceedings of the IEEE/CVF conference on computer vision and pattern recognition. 2022.

---

> ### Author Response · Authors · 2024-11-25
>
> Dear reviewers, thank you for your valuable suggestions! We have answered each reviewer's concerns separately in our response to their comments. If you have any further questions, please feel free to let us know. We look forward to your response and further discussion!

---

> ### Author Response · Authors · 2024-11-29
>
> Happy Thanksgiving, my dear reviewer, I hope you have a happy holiday. If you are satisfied with our work and responses, please consider giving us a higher score. We also welcome your suggestions for our revised manuscripts at any time. Your support is vital to us, thank you!

---

### Official Review · Reviewer_vcVP · 2024-11-02

**Soundness:** 3
**Presentation:** 3
**Contribution:** 2
**Rating:** 5
**Confidence:** 4

**Summary:**

The paper introduces a novel run-time backdoor implantation framework that utilizes protective noise to safeguard sensitive image content in image-editing models, especially against unauthorized modifications. Unlike traditional backdoor techniques, which require training-phase interventions, this approach operates purely during inference, embedding region-specific noise as a backdoor trigger that activates only when sensitive areas are edited. The framework is validated across multiple inpainting models, showing effective resistance to tampering with minimal impact on legitimate editing operations.

**Strengths:**

1. The use of backdoor mechanisms for defensive purposes is novel and reframes backdoor techniques from a security risk to a protective tool, especially in content-sensitive applications.
2. As the method operates without model retraining, it enables the protection of sensitive areas across various pre-trained models with reduced computational requirements, making it feasible for large-scale applications.

**Weaknesses:**

1. The effectiveness of the protective noise may be limited to the specific model for which it is designed, raising concerns about adaptability. The noise generated for one model might not be effective for another due to differences in model architecture and processing.
2. The study does not address whether protection remains effective if an attacker reprocesses the edited image through the same or different models. Multiple edits could potentially bypass the initial protection, which might limit the framework's resilience in real-world applications.

**Questions:**

1. To improve adaptability, the authors could investigate methods to generalize protective noise across different models or make the protective noise more universally applicable.
2. A thorough evaluation of the framework’s resilience to re-editing or re-input scenarios, where the modified image is fed back into the model, would strengthen the practical applicability. Demonstrating protection persistence across multiple editing cycles would address potential bypasses.

---

> ### Author Response · Authors · 2024-11-23
>
> **[Part 1/4]**
>
> **Q1: To improve adaptability, the authors could investigate methods to generalize protective noise across different models or make the protective noise more universally applicable.**
>
> **A1:**  We are grateful for the insightful inquiry posed by the reviewer.
> To assess the cross-model transferability of protective perturbations, we trained images with perturbations based on MAT and then edited them using LaMa. As a result, the protective perturbations still retained a certain resistance effect on LaMa, even though the datasets used for MAT and LaMa training were different. The reverse is not true. We believe that this is because the VIT architecture of MAT has a stronger ability to capture image features than the convolutional network architecture of LaMa, and therefore has a certain degree of transferability. This is also reflected in previous work on similar protective perturbations.
>
> **[Part 2/4]**
>
> Evaluation of cross-model transferability. "MAT.Imp." and "LaMa.Imp." present the implanted image based on MAT and LaMa, "Infer" presents the editing results (inference).
> https://drive.usercontent.google.com/download?id=1NRl3YRxf_ycQLE02jqkfdIJwlf0b-WCZ
> Algorithm:
> https://drive.usercontent.google.com/download?id=1b6cE6kMgwhYtMPZlbRzaJaO_0nFiYfiv
>
> We also discussed two training methods to improve the transferability of different models. One method is parallel backdoor implantation, as shown in Algorithm 1. Taking the optimization of two models at the same time as an example, we add perturbations to the image and input them to the two models at the same time, and optimize the generated results in parallel towards the Target. This method usually requires large computing resources, so we provide a second method - sequential backdoor implantation, as shown in Algorithm 2, which first implants on one model and then fine-tuning the perturbed image on the other model.

---

> ### Author Response · Authors · 2024-11-23
>
> **[Part 3/4]**
>
> **Q2: Demonstrating protection persistence across multiple editing cycles would address potential bypasses.**
>
> **A2:** We thank the reviewer for their professional and meticulous consideration.
> As we describe in the Appendix-A, In image inpainting models, an input is typically provided as an image-mask pair, represented as ⟨x,m⟩. The mask mm is a binary map where the masked areas are represented by the value 1 (indicated visually as white), while the unmasked areas are represented by the value 0 (indicated visually as black). This relationship can be described as follows:
> For a pixel at location (i,j), if the pixel is within the masked region, its value in the mask m(i,j) is 1.
> If the pixel is outside the masked region, its value in the mask m(i,j) is 0.
> This binary mask serves as a guide for the inpainting model by indicating the regions that require prediction and reconstruction (white areas with value 1) and preserving the original content in the unmasked areas (black areas with value 0).
> The final output image is denoted by r, and the predicted content from the inpainting model is represented by G(x). The combination of the predicted content and the original image is governed by the formula:
> r=m⊙G(x)+(1−m)⊙x,
> where ⊙\odot represents element-wise multiplication. This equation indicates that:
> The predicted content from the model, G(x), is applied only to the masked areas (where m=1).
> The original image content, x, is retained in the unmasked areas (where m=0).
> Thus, the final inpainting result r is a composite of the model's prediction in the masked regions and the unchanged original content in the non-masked areas. This process ensures that even if the model's prediction aligns with a specific target, the final output will only reflect the target within the masked region, while the unmasked areas remain intact.
>
> **[Part 4/4]**
> We use LaMa model to show the results of four rounds of continuous editing.
> The qualitative results of experiment:
> https://drive.usercontent.google.com/download?id=1JUFnLQu6iXUaoIb4VRD-vJeVSYjzK9RK.
> In conclusion, the image inpainting process freezes the area outside the mask and restores the content of the masked area based on the pixels of these non-masked areas. Therefore, the surrounding disturbances are rarely changed in this process, which can resist multiple rounds of editing.
> We have revised our manuscript to include these additional discussion and experimental results.

---

> ### Author Response · Authors · 2024-11-25
>
> Dear reviewers, thank you for your valuable suggestions! We have answered each reviewer's concerns separately in our response to their comments. If you have any further questions, please feel free to let us know. We look forward to your response and further discussion!

---

> ### Author Response · Authors · 2024-11-28
>
> Dear reviewer,
>
> Thank you once again for your valuable feedback on my manuscript. We appreciate the time and effort you’ve put into reviewing my work. Thank you for your constructive suggestions, we also have added Appendix G to L in our revised version manuscript.
>
> If you feel that the changes have satisfactorily resolved the issues, We would be grateful if you could consider a higher reassessment of the rating. Your support in this would be greatly appreciated.
>
> Thank you again for your constructive comments and your time.
>
> Best regards, Authors of Submission 6467

---

> ### Author Response · Authors · 2024-11-29
>
> Happy Thanksgiving, my dear reviewer, I hope you have a happy holiday. If you are satisfied with our work and responses, please consider giving us a higher score. We also welcome your suggestions for our revised manuscripts at any time. Your support is vital to us, thank you!

---

### Official Review · Reviewer_T6xT · 2024-11-03

**Soundness:** 3
**Presentation:** 3
**Contribution:** 3
**Rating:** 8
**Confidence:** 4

**Summary:**

This paper presents a runtime backdoor implantation method that protects sensitive image regions from unauthorized edits. By adding subtle protective noise during inference, the approach activates a backdoor only when targeted areas are modified, preventing tampering without retraining the model. Experimental results show effective defense against malicious editing, offering a practical solution for safeguarding image content in various applications.

**Strengths:**

1. Resource-Efficient Without Retraining: The proposed runtime backdoor injection method adds protective noise during inference, eliminating the need for model retraining and reducing resource consumption while preserving the model structure.
2. Region-Aware Trigger, Ensures Legitimate Edits: This method activates the backdoor protection only when specific protected regions are edited, allowing legitimate edits to proceed unaffected and optimizing the loss function to minimize impact on non-trigger regions.
3. Cross-Model Transferability: The approach shows strong transfer performance across various image editing models (e.g., LaMa, MAT, LDM), maintaining effectiveness even with data augmentation techniques like flipping and cropping, demonstrating robustness in diverse applications.

**Weaknesses:**

1. Trade-off between perturbation visibility and anti-editing effectiveness: While increasing the strength of the protective noise improves anti-editing performance, it also introduces noticeable pixel-level changes, which can degrade the visual quality of the image. This trade-off suggests that ensuring strong protection may be challenging without compromising user experience due to the visibility of the perturbation.
2. Dependence on specific trigger regions: The effectiveness of this method relies on designated trigger regions, meaning the backdoor protection only activates when these specific areas are edited. If malicious editing avoids these regions, the protection mechanism may not be effective, limiting the flexibility and applicability of the approach. The region-trigger mechanism also provides attackers with the possibility to evade detection.
3. Limited applicability to diffusion models: Although the method performs well in convolutional and Transformer models, it is less effective in diffusion models (e.g., LDM), especially when the perturbation strength is low, as the protection mechanism may fail. The denoising process in diffusion models tends to neutralize the protective noise, reducing the effectiveness of the approach in these types of models. It is recommended to increase the experiments on other types of generative models (such as GAN or VAE) to further verify the versatility of the method. At the same time, conduct more in-depth experiments on the diffusion model to optimize the perturbation or trigger mechanism and enhance the protection effect.

**Questions:**

1. Will the size of the entire perturbation region affect the effectiveness of the protection? What is the interaction between the noise magnitude and the trigger region? Additional experiments are needed to clarify this.
2. How robust is the proposed solution to various image enhancements (such as brightness adjustment, rotation, and scaling)?

---

> ### Author Response · Authors · 2024-11-23
>
> **[Part 1/6]**
>
> **Q1: Will the size of the entire perturbation region affect the effectiveness of the protection? What is the interaction between the noise magnitude and the trigger region? Additional experiments are needed to clarify this.**
>
> **A1:** We are grateful for the insightful inquiry posed by the reviewer. In our method, we use perturbations of the same shape as the image to achieve protection. This is because we not only want to resist edits to sensitive areas, but also want to legitimize benign edits to non-sensitive areas. To better clarity this, we have written this part in section 3.2 of our revised version.
> To explore the interaction between the noise magnitude and trigger region, we calculate the average pixel difference between the perturbed image and the original image in the trigger area, as well as the average pixel difference in the non-trigger area, and the results are shown in the Table R1-1.
>
> **[Part 2/6]**
>
> **Table R1-1: pixel difference in the trigger/non-trigger area between perturbed image and original image .**
>
> |  | **LDM** |  | **LaMa** |  | **MAT** |  |
>
> |**Perturbation Bound**|Non-trigger|Trigger|Non-trigger|Trigger|Non-trigger|Trigger|
>
> | **6/255** | 1.906e-2 | 0 | 2.22e-2 | 0 | 2.055e-2 | 0 |
>
> | **13/255** | 3.543e-2 | 0 | 3.875e-2 | 0 | 3.86e-2 | 0 |
>
> As we describe in the Appendix-A, In image inpainting models, an input is typically provided as an image-mask pair, represented as ⟨x,m⟩. The mask mm is a binary map where the masked areas are represented by the value 1 (indicated visually as white), while the unmasked areas are represented by the value 0 (indicated visually as black). This relationship can be described as follows:
> For a pixel at location (i,j), if the pixel is within the masked region, its value in the mask m(i,j) is 1.
> If the pixel is outside the masked region, its value in the mask m(i,j) is 0.
> This binary mask serves as a guide for the inpainting model by indicating the regions that require prediction and reconstruction (white areas with value 1) and preserving the original content in the unmasked areas (black areas with value 0).
> The final output image is denoted by r, and the predicted content from the inpainting model is represented by G(x). The combination of the predicted content and the original image is governed by the formula:
> r=m⊙G(x)+(1−m)⊙x,
> where ⊙\odot represents element-wise multiplication. This equation indicates that:
> The predicted content from the model, G(x), is applied only to the masked areas (where m=1).
> The original image content, x, is retained in the unmasked areas (where m=0).
> The final inpainting result r is a composite of the model's prediction in the masked regions and the unchanged original content in the non-masked areas. This process ensures that even if the model's prediction aligns with a specific target, the final output will only reflect the target within the masked region, while the unmasked areas remain intact.
> Thus, the perturbations are concentrated around the trigger area, while the pixel distribution in the trigger area is basically the same as that in the original image.

---

> ### Author Response · Authors · 2024-11-23
>
> **[Part 3/6]**
>
> **Q2: How robust is the proposed solution to various image enhancements (such as brightness adjustment, rotation, and scaling)?**
>
> **A2:** We express our gratitude to your insightful question. We add an experiment to explore the robustness of our perturbations to image enhancement operations.
> In our experiments, we evaluated the robustness of the proposed attack under several common image transformations. The results demonstrate that the solution remains effective even after the application of standard image enhancements. Specifically, we tested brightness adjustment (increased by a factor of 1.5), scaling (doubling the image size), blurring (with a blur radius of 3), a 90-degree counterclockwise rotation, and horizontal flipping. In all cases, the attack continued to succeed, indicating that the proposed solution is robust to variations in resolution, rotation, and image reflections.
> In the case of blurring, our attack method is destructive to the image in some degree, as blur itself degrades the image quality to an unusable level, indicating that such transformations have a significant impact on image usability regardless of whether or not it is attacked.
> These findings underscore the resilience of the attack to typical image augmentations and suggest that the solution is robust to common preprocessing operations to some extent.
>
>
> **[Part 4/6]**
> The qualitative results of experiment:
> https://drive.usercontent.google.com/download?id=15q2ZhvJtD7v_Gpn97RMggZeM1HJpYMUO.
>
> We have revised our manuscript to include these additional discussion and experimental results.
>
>
> **[Part 5/6]**
>
> **Weakness: conduct more in-depth experiments on the diffusion model to optimize the perturbation or trigger mechanism and enhance the protection effect. **
>
> **A3:** We found an approach to compare with our work that is closer to the scene (also introduced in the related work of our original paper), namely image immunization ([1]ICML 2023 Oral), which has two paradigms: encoding attack and diffusion attack. The encoder attack aims to attack the encoder of the diffusion model, causing the diffusion model to receive inferior image embeddings. The diffusion attack aims to attack the entire diffusion process, obtaining better resistance at a larger computational cost. We provide comparison results for both paradigms and supplement them in the revised version.  We use the same model as them, the latent diffusion model, and the comparison dataset is CelebA-HQ. As shown in Table R1-2, our method is still the best compared to the two paradigms with additional baseline.
> And in Table R1-3, We analyze the performance of using different perturbation bounds to resist the diffusion model editing.
> We have revised our manuscript to include these additional discussion and experimental results.

---

> ### Author Response · Authors · 2024-11-23
>
> **[Part 6/6]**
>
> **Table R1-2: Attack performance comparison with additional baselines (image immunization).**
> | **Method** | **CLIP-FID** | **LPIPS** | **SSIM** | **PSNR** |
> |:---:|:---:|:---:|:---:|:---:|
> | **Benign** | 18.28 | 0.047 | 0.316 | 13.94 |
> | **Encoder Attack [1]** | 27.85 | 0.069 | 0.181 | 12.20 |
> | **Encoder Backdoor (ours)** | 28.94 | 0.075 | 0.156 | 10.98 |
> | **Diffusion Attack [1]** | 35.56 | 0.096 | 0.123 | 7.28 |
> | **Runtime Backdoor (ours)** | 40.55 | 0.101 | 0.097 | 7.20 |
> Whether attacking only the encoder of diffusion model or the entire diffusion process, our method can effectively resist image editing and achieve SOTA performance.
>
>
> **Table R1-3: Ablation on Perturbation Bound.**
> | **Method** | **CLIP-FID** | **LPIPS** | **SSIM** | **PSNR** |
> |:---:|:---:|:---:|:---:|:---:|
> | **Benign** | 18.28 | 0.047 | 0.316 | 13.94 |
> | **3/255** | 19.53 | 0.048 | 0.262 | 13.11 |
> | **6/255** | 40.55 | 0.101 | 0.097 | 7.20 |
> | **13/255** | 36.26 | 0.128 | 0.110 | 5.29 |
>
> Our method can significantly affect the latent diffusion model with a perturbation of 6/255. There are two points worth noting here. First, the diffusion model itself has a denoising function, which will weaken the perturbation to a certain extent. Second, when the perturbation bound increases from 6/255 to 13/255, the CLIP-FID in the image editing results remains comparable is close, with the latter not consistently showing improvement (this is consistent with what we described in section 4.3). To get a better protection, we still recommend appropriately increasing the perturbation bound for the diffusion model.
>
> Text-guided inpainting uses an image-mask-text triplet, where the missing region is filled based on both the surrounding image context and the textual input. In particular, text will be used to generate content to fill the mask area. Pixels outside the mask area are only used to smooth the content of text generation. In particular, text will be used to generate content to fill the mask area. Pixels outside the mask area are only used to smooth the content of text generation. Therefore, when using this method to resist the method of text guidance, performance is limited.
> We use Stable Diffusion for the editor of text guidance to evaluate our methods. The experimental results are shown below. Although it is difficult for us to intercept the production of text content, protective disturbances can still bring certain distortion to the generated images, such as making pixel colors close to target. The results of the qualitative analysis are shown in the figure：https://drive.usercontent.google.com/download?id=1btJ6ThtdcgXVv2EbFpHWHOhelBomS-BH
> The prompt we use is "Tail of a dog, high resolution, sitting on a bench in the park."
>
> **References:**
>
> [1] Salman, Hadi, et al. "Raising the cost of malicious AI-powered image editing." Proceedings of the 40th International Conference on Machine Learning. 2023. (Oral)

---

> ### Author Response · Authors · 2024-11-25
>
> Dear reviewers, thank you for your valuable suggestions! We have answered each reviewer's concerns separately in our response to their comments. If you have any further questions, please feel free to let us know. We look forward to your response and further discussion!

---

> ### Comment · Reviewer_T6xT · 2024-11-26
>
> I have seen your reply and my problem has been solved.

---

> > ### Author Response · Authors · 2024-11-26
> >
> > Thank you again for your time and constructive suggestions! :-)

---

### Official Review · Reviewer_hkNz · 2024-11-09

**Soundness:** 3
**Presentation:** 3
**Contribution:** 3
**Rating:** 5
**Confidence:** 4

**Summary:**

This paper presents a method for image content protection by implanting run-time backdoors in image-editing models and triggering the backdoors when sensitive content on an image is modified by an image-editing model. The triggered backdoor will then stop the editing from happening. The backdoors are injected via imperceptible perturbations on the images and the protected sensitive area in an image can be defined to trigger the backdoor. Experiments are performed on several images and evaluated using standard image similarity metrics.

**Strengths:**

The method of implanting run-time backdoors to prevent editing of sensitive image content is interesting.

The method, technique and experiments are also presented well.

**Weaknesses:**

It is not clear from the paper how robust the method is on image distortions and perturbations such as compression, smoothening, even before editing of sensitive image content is performed. In a realistic scenario, an image can go through such distortions. It is also not clear from the paper, how the method will respond to global image transformations such as blur.

**Questions:**

What are the limitations of the proposed approach? If an adversary is aware of this approach, will the adversary be able to evade?

---

> ### Author Response · Authors · 2024-11-23
>
> **[Part 1/4]**
>
> **Q1: What are the limitations of the proposed approach?**
>
> **A1:** We are grateful for the reviewer's constructive suggestion. In Appendix G (Discussion on Limitations) of original paper, we have discussed the limitations of the proposed method.
> Besides, based on the reviewers’ comments, we have further analyzed and discussed the results, and the detailed results are shown in Appendix G of revised version.
> “Our proposed method introduces the first run-time backdoor implantion framework that eliminates the need for model retraining. As a result, we prioritize simple and effective backdoor targets over more complex ones. This choice arises from the observation that the similarity between the generated outputs and the intended target is influenced not only by the introduced perturbations but also by the inherent generative capacity of the model. In our attack-as-defense scenario, the generative ability of the model is not manipulated by the backdoor implanter(defender).
>
> Furthermore, our approach targets open-source white-box models, which implies that the computational resources required depend heavily on the forward computation process of the model, making it difficult to quantify. Nevertheless, the method itself only introduces parameters proportional to the size of the image (we have already analyzed computational resources in Appendix F). Future work could explore strategies such as reinforcement learning or greedy search to implement attack-as-defense in black-box models, which would make the computational resource requirements more quantifiable.”
>
>
> **[Part 2/4]**
>
> **Q2: If an adversary is aware of this approach, will the adversary be able to evade?**
>
> **A2:** We thank the reviewer for their professional and meticulous consideration. Here we discuss this in two cases: first, can adversary predict whether the image is protected? Second, can adversary destroy our protective perturbation through global image transformations?
>
> For the first case. In our Threat Model, we consider whether an adversary can detect whether the image is protected or original before editing. We believe that even if the adversary knows the existence of our method, it is still difficult to detect it in advance. Because our method is designed to make the copyright holder of image publish protected images instead of publishing the original images, the adversary can only get the protected images, she/he can only fail to edit or be forced to change the image editing tool, which also increases the cost of maliciously editing the images. Similar threat models can refer to existing work ([1]Salman, Hadi, et al. ICML 2023 (oral) ).
>
> For the second case. We add an experiment to verify the robustness of the proposed method to global image transformations. In our experiments, we evaluated the robustness of the proposed attack under several common image transformations. The results in Figure of [Part 3/4] demonstrate that the solution remains effective even after the application of standard image enhancements. Specifically, we tested brightness adjustment (increased by a factor of 1.5), scaling (doubling the image size), blurring (with a blur radius of 3), a 90-degree counterclockwise rotation, and horizontal flipping. In all cases, the attack continued to succeed, indicating that the proposed solution is robust to variations in resolution, rotation, and image reflections. In the case of blurring, our attack method is destructive to the image in some degree, as blur itself degrades the image quality to an unusable level, indicating that such transformations have a significant impact on image usability regardless of whether or not it is attacked.These findings underscore the resilience of the attack to typical image augmentations and suggest that the solution is robust to common preprocessing operations to some extent.
>
> **[Part 3/4]**
>  The qualitative results of experiment:
> https://drive.usercontent.google.com/download?id=15q2ZhvJtD7v_Gpn97RMggZeM1HJpYMUO.
>
>
>
> **[Part 4/4]**
>
> **Q3: How the method will respond to global image transformations such as blur.**
>
> **A3:** We are grateful for the reviewer's insightful question. We have discussed the robustness of various image transformations in Q2.
>
>
>
> **References:**
>
> [1] Salman, Hadi, et al. "Raising the cost of malicious AI-powered image editing." Proceedings of the 40th International Conference on Machine Learning. 2023. (Oral)

---

> ### Author Response · Authors · 2024-11-25
>
> Dear reviewers, thank you for your valuable suggestions! We have answered each reviewer's concerns separately in our response to their comments. If you have any further questions, please feel free to let us know. We look forward to your response and further discussion!

---

> ### Comment · Reviewer_hkNz · 2024-11-27
>
> Thanks to the authors for addressing my questions and queries. I find them satisfactory.

---

> > ### Author Response · Authors · 2024-11-27
> >
> > Dear reviewer,
> >
> > Thank you once again for your valuable feedback on my manuscript. I appreciate the time and effort you’ve put into reviewing my work. Thank you for your constructive suggestions, we also have added Appendix G to L in our revised version manuscript.
> >
> > If you feel that the changes have satisfactorily resolved the issues, I would be grateful if you could consider a higher reassessment of the rating. Your support in this would be greatly appreciated.
> >
> > Thank you again for your constructive comments and your time.
> >
> > Best regards,
> > Authors of Submission 6467

---

> > ### Author Response · Authors · 2024-11-29
> >
> > Happy Thanksgiving, my dear reviewer. We hope you have a happy holiday. If you are satisfied with our work and responses, please consider giving us a higher score. We also welcome your suggestions for our revised manuscripts at any time. Your support is very important to us, thank you!

---

### Author Response · Authors · 2024-11-23
**General Response to All Reviewers**

We thank all the reviewers for their insightful questions and constructive suggestions! We are glad that the reviewers found our paper “method is interesting”, “technique and experiments are presented well”, “strong transfer performance”,“a novel  framework”, “feasible for large-scale applications”, “effectively”, “propose an innovative and practical approach”, “valuable both academically and practically”, “demonstrates strong experimental results”, and “providing a useful approach”.

We have addressed each reviewer's concerns individually in our responses to their reviews. Should you have any further questions, please do not hesitate to let us know. We are fully prepared to address any additional concerns that may arise.

---

### Comment · Area_Chair_pFH5 · 2024-11-23

Dear Reviewers,
The authors have responded to your valuable comments.
Please take a look at them!

Best,
AC

---

### Author Response · Authors · 2024-11-24

Dear reviewers, thank you for your valuable suggestions! We have answered each reviewer's concerns separately in our response to their comments. If you have any further questions, please feel free to let us know. We look forward to your response and further discussion!

---

### Author Response · Authors · 2024-11-25

Dear reviewers, we have added Appendix G-L in our revised version manuscript. Please have a look! Looking forward to having further discussion with you.

---

### Meta-Review · Area_Chair_pFH5 · 2024-12-17

**Metareview:**

In this paper, the authors proposed a run-time backdoor attack to prevent the sensitive region from being tampered with.
There are several comments raised by the reviewers, so the authors replied to 18 questions raised by the reviewers, provided additional 7 experiments, and included a detailed appendix (from G to L) in the revised version.
The most critical question that the authors did not respond well is the robustness evaluation; that is, what kinds of editing/distortions that the embedded backdoor can resist?
For image protection via backdooring/watermarking, robustness is a critical requirement that needs sufficient evaluation.

**Additional Comments On Reviewer Discussion:**

During the rebuttal period, only the Reviewer hkNz responded to authors' responses!
However, the authors did not respond to the comment, ``It is not clear from the paper how robust the method is on image distortions and perturbations such as compression, smoothening, even before editing of sensitive image content is performed,'' made by Reviewer hkNz.
The AC checked the results provided by the authors but cannot find any results relevant to resistance against compression.

---

### Decision · Program_Chairs · 2025-01-22

Reject